# The APOBEC3B cytidine deaminase is an adenovirus restriction factor

**Noémie Lejeune[1]***, **Sarah Mathieu[1]**, **Alexandra Decloux[1]**, **Florian Poulain[1]**, **Zoé Blockx[1]**, **Kyle A. Raymond[2,3]**, **Kévin Willemart[1]**, **Jean-Pierre Vartanian[2]**, **Rodolphe Suspène[2]**, **Nicolas A. Gillet** [1]*

**1** Namur Research Institute for Life Sciences (NARILIS), Integrated Veterinary Research Unit (URVI), University of Namur, Namur, Belgium, **2** Département de Virologie, Institut Pasteur, Paris, France, **3** Sorbonne Université, Collège Doctoral, Paris, France

* lejeune.noemie@gmail.com (NL); nicolas.gillet@unamur.be (NAG)

## Abstract

Human adenoviruses (HAdVs) are a large family of DNA viruses counting more than a hundred strains divided into seven species (A to G). HAdVs induce respiratory tract infections, gastroenteritis and conjunctivitis. APOBEC3B is a cytidine deaminase that restricts several DNA viruses. APOBEC3B is also implicated in numerous cancers where it is responsible for the introduction of clustered mutations into the cellular genome. In this study, we demonstrate that APOBEC3B is an adenovirus restriction factor acting through a deaminase-dependent mechanism. APOBEC3B introduces C-to-T clustered mutations into the adenovirus genome. APOBEC3B reduces the propagation of adenoviruses by limiting viral genome replication, progression to late phase, and production of infectious virions. APOBEC3B restriction efficiency varies between adenoviral strains, the A12 strain being more sensitive to APOBEC3B than the B3 or C2 strains. In A12-infected cells, APOBEC3B clusters in the viral replication centers. Importantly, we show that adenovirus infection leads to a reduction of the quantity and/or enzymatic activity of the APOBEC3B protein depending on the strains. The A12 strain seems less able to resist APOBEC3B than the B3 or C2 strains, a characteristic which could explain the strong depletion of the APOBEC3-targeted motifs in the A12 genome. These findings suggest that adenoviruses evolved different mechanisms to antagonize APOBEC3B. Elucidating these mechanisms could benefit the design of cancer treatments. This study also identifies adenoviruses as triggers of the APOBEC3B-mediated innate response. The involvement of certain adenoviral strains in the genesis of the APOBEC3 mutational signature observed in tumors deserves further study.

## Author summary

The APOBEC3B protein is overexpressed in many human tumors and promotes genetic diversity and resistance to drug and oncolytic viruses. The identification of the mechanisms that lead to the dysregulation of APOBEC3B and the search for ways to abrogate its expression in tumors are intensively pursued. We have previously shown that adenoviral infection triggers APOBEC3B expression. In this study, we demonstrate that APOBEC3B

**Data Availability Statement:** All relevant data are within the manuscript and its Supporting Information files.

**Funding:** This study was supported by FRS-FNRS grant CDR n°31270116 and n°40007814 and by the

University of Namur. NL is a PhD fellow supported by FRIA grant n°31454280. SM is a PhD fellow supported by FRIA grant n°40008651. FP is a PhD fellow supported by Télévie grant PDR-TLV n° 34972507. AD is a PhD fellow supported by an FSR grant co-founded between University of Namur (Belgium) and University of Mons (Belgium). KAR is supported by the Allocation de Recherche du Ministère français de la Recherche. The funders had no role in study design, data collection and interpretation, or the decision to submit the work for publication.

**Competing interests:** The authors have declared that no competing interests exist.

mutates the genome of adenoviruses during their replication in the nucleus. These results warrant further investigation of the role of adenoviruses in APOBEC3B dysregulation. Our study also suggests that adenoviruses have evolved mechanisms to antagonize APOBEC3B and that some adenoviral strains are highly capable of decreasing the amount of APOBEC3B protein in infected cells. Identification of the viral proteins involved could be used to design better oncolytic vectors capable of propagating in tumor cells expressing high levels of APOBEC3B.

## Introduction

Human adenoviruses (HAdVs) are non-enveloped double-stranded DNA viruses that replicate their linear genome in the nucleus. HAdVs cause upper and lower respiratory tract, ocular, and gastrointestinal infections [1]. HAdVs are very diverse, counting more than a hundred genotypes divided among 7 species (A to G) [2]. The most common genotypes detected differ among countries and change over time [3]. In the United States, HAdV-C1, -C2, and -B3 are the three most frequently reported genotypes [4]. Although generally self-limiting in adults, HAdVs can lead to acute and lethal infection in early childhood and immunocompromised individuals.

The APOBEC3 enzymes (A3s) are innate antiviral effectors (reviewed in [5,6]). The human genome encodes seven A3 genes: APOBEC3A, B, C, D, F, G and H. The A3s are cytidine deaminases that bind the viral DNA, converting cytidine to uridine. The A3s generally deaminate cytidine in a 5'-TC motif, with the exception of A3G that favors editing of cytidine when preceded by another cytidine (*i.e.* in a 5'-CC motif). The A3s are processive and introduce strand-coordinated clustered C to U mutations, a phenomenon called hyperediting or hypermutation. The hypermutated viral sequences are subsequently rendered nonfunctional. The A3 proteins also exert an antiviral activity through a deaminase-independent mechanism, by impeding the polymerization of the viral genome. The APOBEC3B (or A3B) protein has been shown to restrict the reverse-transcribing viruses hepatitis B virus (HBV), human immunodeficiency viruses (HIVs) and human T lymphotropic virus-1 (HTLV-1) [7–10], endogenous retroelements [11,12], as well as the double-stranded DNA viruses Epstein-Barr virus (EBV), Kaposi's sarcoma-associated herpesvirus (KSHV) and herpes simplex virus-1 (HSV-1) [13,14]. Not surprisingly, these viruses have evolved different mechanisms to counteract A3B. For instance, EBV, KSHV and HSV-1 evade A3B by a combination of A3B subcellular relocation and/or inactivation of its enzymatic activity [13,14], whereas HIV-2 reduces A3B cellular level and inhibits its packaging into virions [7]. In a less expected way, it has been demonstrated that the small double-stranded DNA viruses human papillomaviruses (HPVs) and BK/JC polyomaviruses (PyVs) promote the expression of A3B via their oncoproteins E6 and Large T antigen [15–18]. A3B has been proposed to increase the genetic diversity of these viruses and promote immune escape [19,20].

We demonstrated that adenovirus genomes were depleted for the 5'TC motif that is favored by most of the A3s [21]. We have also shown that adenovirus infection of bronchial cells leads to an increase in A3B transcription [22]. However, at the protein level, A3B is abundantly expressed during the infection of bronchial cells with the A12 strain, yet less so during HAdV-C2 infection, and almost undetectable in HAdV-B3-infected cells. In this study, we examined the impact of A3B on the replication of adenoviruses. The B3 and C2 strains were chosen based on their high incidence, and the A12 for its particular ability to induce tumors in animal models [23–25]. Our results demonstrate that A3B is an adenovirus restriction factor and that adenoviruses have evolved to counteract its action.

## Results

### A3B limits the propagation of adenoviruses by a deaminase-dependent mechanism

The human epithelial bronchial cells (HBEC3-KT, thereafter named HBEC-WT for Wild Type) were transduced by lentiviral vectors in order to induce a constitutive expression of A3B. Three different cell lines were established: the HBEC-A3B cells (expressing an enzymatically active A3B protein), the HBEC-A3B-DD cells (producing a Deaminase Dead A3B mutant) and the HBEC-GFP cells as control. The levels of A3B mRNA in the HBEC-A3B and HBEC-A3B-DD cells are increased by more than 100 folds compared to WT, whereas GFP expression does not impact A3B mRNA level (Fig 1A). The HBEC-A3B and A3B-DD express high and comparable levels of A3B protein (Fig 1B, lanes 3 and 4) but the deaminase activity is only evident in the HBEC-A3B cells (Fig 1C, lane 3). Of note, S1 Fig shows the levels of A3B mRNA and protein expressed in HBEC-A3B cells compared to those observed in HBEC-WT infected with the different adenoviral strains.

We infected HBEC-WT, HBEC-A3B, HBEC-A3B-DD and HBEC-GFP with the different adenoviral strains, A12, B3 and C2 at a low MOI (0.03), and monitored viral replication in the 4 cell lines. The experiments were conducted in three biological replicates, each replicate being independent (i.e. cells plated and infected on different days). Fig 1D shows the percentage of infected cells measured by flow cytometry at different time points post infection. We first observed that the propagation of the A12 strain is consistently slower than that of the B3 and C2 strains. For this reason, we extend the follow-up of the A12 infection up to 10 days post infection (dpi), whereas we stopped our analysis after 7 days for the B3 and C2 strains. We also observed that viral propagation is repeatedly slower in HBEC-A3B cells (red line) than in HBEC-WT, HBEC-GFP or HBEC-A3B-DD (respectively yellow, green and black lines). It should be noted that the propagation of the B3 strain during the third replicate was faster than during the first two. Most of the cells were lysed at 7 days post-infection, precluding FACS and intracellular DNA qPCR analysis for this time point. To allow a summarized representation and a statistical analysis of the data, Fig 1E depicts the percentage of infected cells measured during the 3 replicates. We observed that the percentage of A12-infected cells in A3B-expressing cells is significantly lower compared to the percentage measured in the WT cells and that from 2 days post infection and up to 10 days (Fig 1E, left panel, red bars versus yellow bars). On the contrary, the percentage of A12-infected cells in GFP or A3B-DD cells is not different to the percentage measured in the WT cells (Fig 1E, left panel, green and black bars versus yellow bars). We conclude that A3B restricts the propagation of the A12 virus by a deaminase-dependent mechanism. The percentage of B3-infected cells in A3B-expressing cells is significantly lower compared to the percentage measured in the WT cells at 2 days post infection but no significant difference could be observed later on (Fig 1E, middle panel, red bars). Finally, the percentage of C2-infected cells in A3B-expressing cells is significantly lower compared to the percentage measured in the WT cells at 2- and 4- days post infection but no significant difference could be observed at 7 dpi (Fig 1E, right panel, red bars). We conclude that A3B limits the propagation of B3 and C2 but with less efficiency than against A12 strain.

Fig 2A illustrates the intracellular viral load detected in the cells by qPCR at different time-points post-infection. Data from the 3 replicates are shown in S2 Fig. We first observed that the viral load in HBEC-GFP, -A3B or -A3B-DD cells measured at 6 hours post infection is not different from the viral load measured in WT cells (Fig 2A, 6 hpi, red, green and black bars versus yellow bars). We conclude that the expression of A3B or A3B-DD does not impact virus entry. We observed that the viral load in A3B-expressing cells infected with the A12 strain is significantly lower compared to the load measured in WT cells and that from 24 hours post

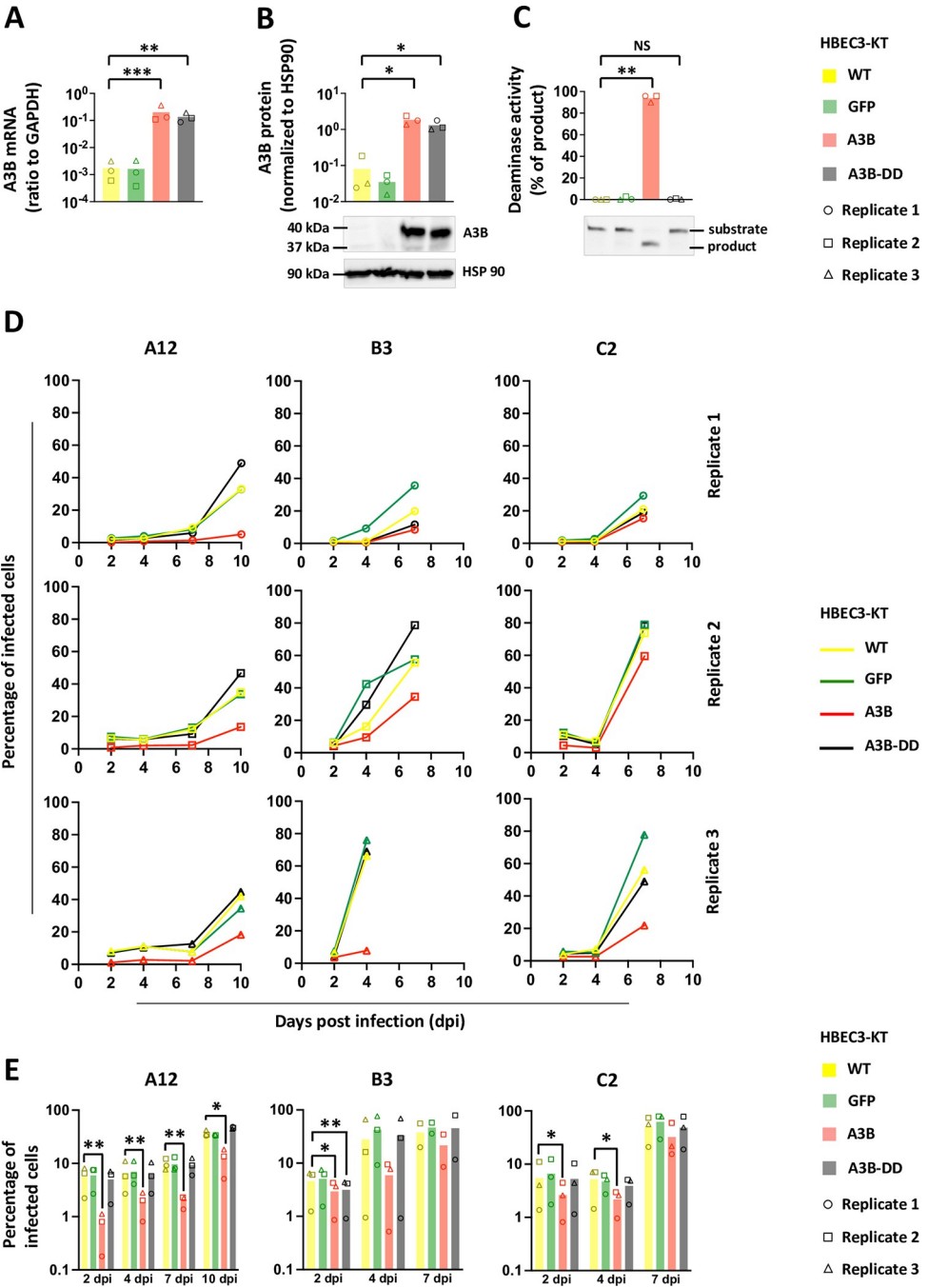

**Fig 1. A3B slows down adenovirus propagation by a deaminase-dependent mechanism.** Immortalized human bronchial epithelial cells HBEC3-KT were transduced to stably express the GFP, the APOBEC3B protein (A3B) or a catalytically inactive version of the APOBEC3B protein (A3B-DD for Deaminase Dead). (A) A3B mRNA levels were quantified by RTqPCR and expressed relative to GAPDH mRNA level. (B) A3B protein levels were assessed by western blot, quantified by densitometry and normalized with HSP90. (C) Total proteins were extracted in a non-denaturing buffer and mixed with the substrate of the deamination assay. Deaminase activity, if present in the cell lysate, will allow the conversion of substrates into shorter products. Substrates and products were quantified by densitometry. Deamination activities of the cell lysates were expressed as the percentage of substrate converted into product. P-values were calculated by t-tests. (D-E) HBEC-WT, -GFP, -A3B and -A3B-DD were infected with HAdV-A12, -B3 or -C2 at a MOI = 0.03. The percentage of infected cells were quantified by flow cytometry at different time points post infection. The results of the 3 biological replicates are shown separately (D) or pooled together (E). P-values were calculated by ratio t-tests. Only significant differences are shown.

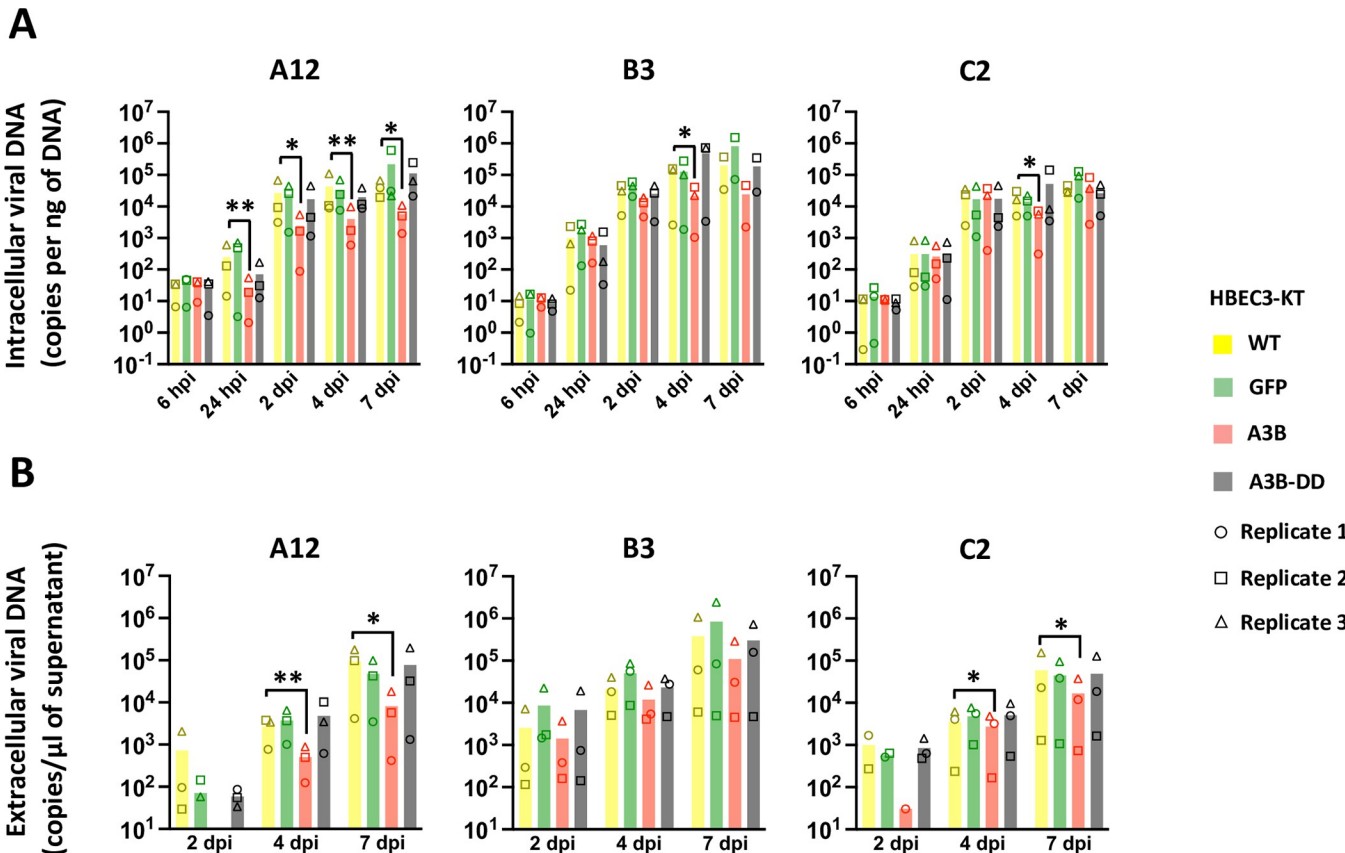

**Fig 2. A3B slows down viral DNA production.** HBEC-WT, -GFP, -A3B and -A3B-DD were infected with HAdV-A12, -B3 or -C2 at a MOI = 0.03. Intracellular viral DNA levels (A) and extracellular viral DNA levels (B) were quantified by qPCR at different time points post infection. P-values were calculated by ratio t-tests. Only significant differences are shown.

infection and up to 7 days (Fig 2A, left panel, red bars). The viral load in A3B-expressing cells infected with the B3 or the C2 strain is lower compared to the load measured in WT only at 4 days post infection (Fig 2A, middle and right panels). No difference can be observed between the GFP or A3B-DD cells and the WT cells. Fig 2B illustrates the extracellular viral load detected in the supernatant by qPCR at different time-points post-infection. Data from the 3 replicates are shown in S3 Fig. The viral load in the supernatant of A3B-expressing cells infected with the A12 or C2 strains is lower compared to the load measured in WT cells at 4 and 7 dpi (Fig 2B, left and right panels). No difference of viral load in the supernatant was observed in cells infected with the B3 strain (Fig 2B, middle panel).

Overall, we conclude that A3B restricts the replication of these different adenoviral strains, that the deaminase activity of the protein is required, and that the magnitude of restriction is influenced by the strain of virus. The most sensitive strain being the A12, followed by the C2 and finally the B3 which is only marginally impacted by the A3B protein.

## A3B reduces viral load, progression to late phase transcription, and production of infectious viral particles with variable efficacy between viral strains

To better characterize the mechanism of A3B restriction, we infected the HBEC-WT, -GFP, -A3B and -A3B-DD with the 3 strains at a high MOI (MOI = 3) to get a synchronous infection.

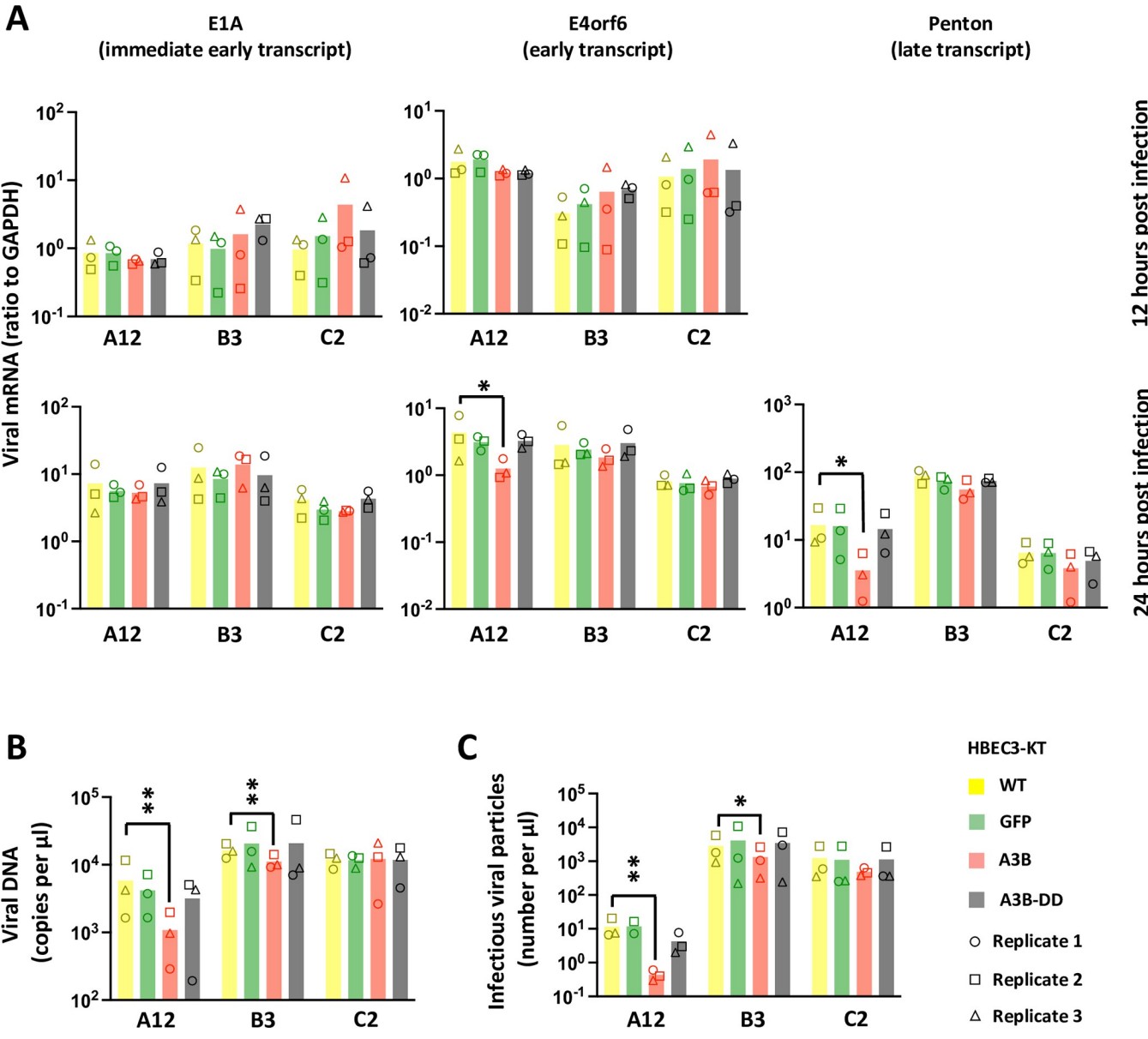

**Fig 3. A3B reduces viral load, progression to late phase transcription, and production of infectious viral particles with variable efficacy between viral strains.** HBEC-WT, -GFP, -A3B and -A3B-DD were infected with HAdV-A12, -B3 or -C2 at a MOI = 3. (A) The levels of different viral mRNAs were quantified by RTqPCR after 12 and 24-hours post infection. An immediate early transcript (E1A), an early transcript (E4orf6) and a late transcript (Penton) were quantified. The penton mRNA could not be reliably detected at 12-hours post infection. (B) The levels of viral DNA were quantified by qPCR at 24 hours post infection. (C) The levels of infectious viral particles were quantified by foci forming assay at 24 hours post infection. P-values were calculated by ratio t-tests. Only significant differences are shown.

We measured the expression of immediate early, early and late transcripts by RTqPCR at 12 and 24-hours post infection. At 24-hours post infection, we measured the replication of the viral DNA by qPCR and the production of infectious viral particles by foci forming assays. We observed that the expression of the immediate early gene E1A is not impacted in any conditions (Fig 3A, left histograms), indicating that the virus enters the cell and starts its replication program regardless of the A3B status of the infected cell. At 24-hours post infection, we observed a decrease in the abundance of the early transcript E4orf6 and the late transcript

Penton in the HBEC-A3B cells infected with the A12 strain compared to WT cells (Fig 3A, bottom middle and right histograms, red bars vs yellow bars). Fig 3B shows that viral DNA load is lower in cells expressing the enzymatically active A3B protein in the A12- and B3-infected conditions. Finally, virions have been isolated for each condition and their infectivity has been measured by a foci forming assay. Fig 3C shows that the A12- and B3-infected cells expressing the enzymatically active A3B protein produce a lower amount of infectious viral particles than their WT counterpart. The lower production of infectious viral particles in the HBEC-A3B cells (Fig 3C) can be explained by the lower viral DNA load (Fig 3B).

We conclude that A3B does not impact the entry of the virus, nor the expression of immediate early genes. Rather, A3B decreases viral DNA replication and delays the progression into the late phase, resulting in a lower production of infectious viral particles.

## APOBEC3B introduces strand-coordinated clustered C to T mutations the viral genome of the A12, B3 and C2 strains

The results obtained so far show that A3B restricts adenovirus replication in a deaminase-dependent way and via the reduction of the viral genome replication. We therefore wonder whether A3B hypermutates the viral DNA. HBEC-WT, -GFP, -A3B and -A3B-DD were infected with HAdV-A12, -B3 or -C2 and the intracellular DNA was extracted 48 hours post infection. The selective amplification of viral sequences carrying strand-coordinated clustered C to T mutations was performed by 3DPCR (Differential DNA Denaturation). We have previously reported that adenoviruses bear an A3 evolutionary footprint [21,22]. We designed 3DPCR reactions on 3 different sections of the viral genome: at the ends (E1B and E4 genes) and in the middle (L3 gene) of the linear genome. Fig 4A illustrates the E4-specific 3DPCR amplicons obtained during HAdV-A12 infection of WT-, GFP-, A3B- and A3B-DD-expressing cells. We observed low denaturation temperature amplicons (Fig 4A, third gel from the top, white stars) in the 3DPCR reactions done on the DNA extracted from A3B-expressing cells but not from the other cell lines. Fig 4B reports the 3DPCR results for the 3 viral genes (E1B, L3 and E4), the 3 viral strains (A12, B3 and C2) and the 4 cell lines (WT, GFP, A3B and A3B-DD). We observed low denaturation temperature amplicons in A3B-expressing cells, but not in the cells expressing the catalytically inactive form of A3B, nor in the WT or GFP cells. Importantly, we observed low denaturation temperature amplicons in A3B-expressing cells for the 3 strains. We cloned and sequenced amplicons obtained during HAdV-A12, -B3 and -C2 infection of WT and A3B-expressing cells (Figs 4C and S4). Fig 4C reports representative sequences for the E1B, L3 and E4 genes isolated from WT or A3B-expressing cells infected with the A12 strain. The sequences from WT cells were not different from the reference genome. We also reported 2 different and representative hypermutated sequences for each gene. Deamination events were observed in both the coding (C to T mutations) and template strand (C to T mutations in the template strand resulting in G to A mutations in the coding strand) of the viral sequences. As expected, when we recover a hypermutated sequence, the mutations are almost exclusively C to T (mutations of the coding strand) or G to A (C to T mutations of the template strand). We have never observed a sequence with both C to T and G to A clustered mutations. However, we have observed mutation clusters on both strands of the same gene. For instance, for the L3 gene, we found 3DPCR amplicons with C to T mutations and other amplicons with G to A mutations (S4 Fig). The mutations translate into synonymous and non-synonymous substitutions and premature stop codons (S4 Fig). Fig 4D shows that the mutations recorded in the hypermutated clones are almost exclusively C to T or G to A (making more than 97% of the mutations). Lastly, we looked at the nucleotide context upstream of the mutated C. Fig 4E shows an enrichment for a T in 5' of the mutated C, both in

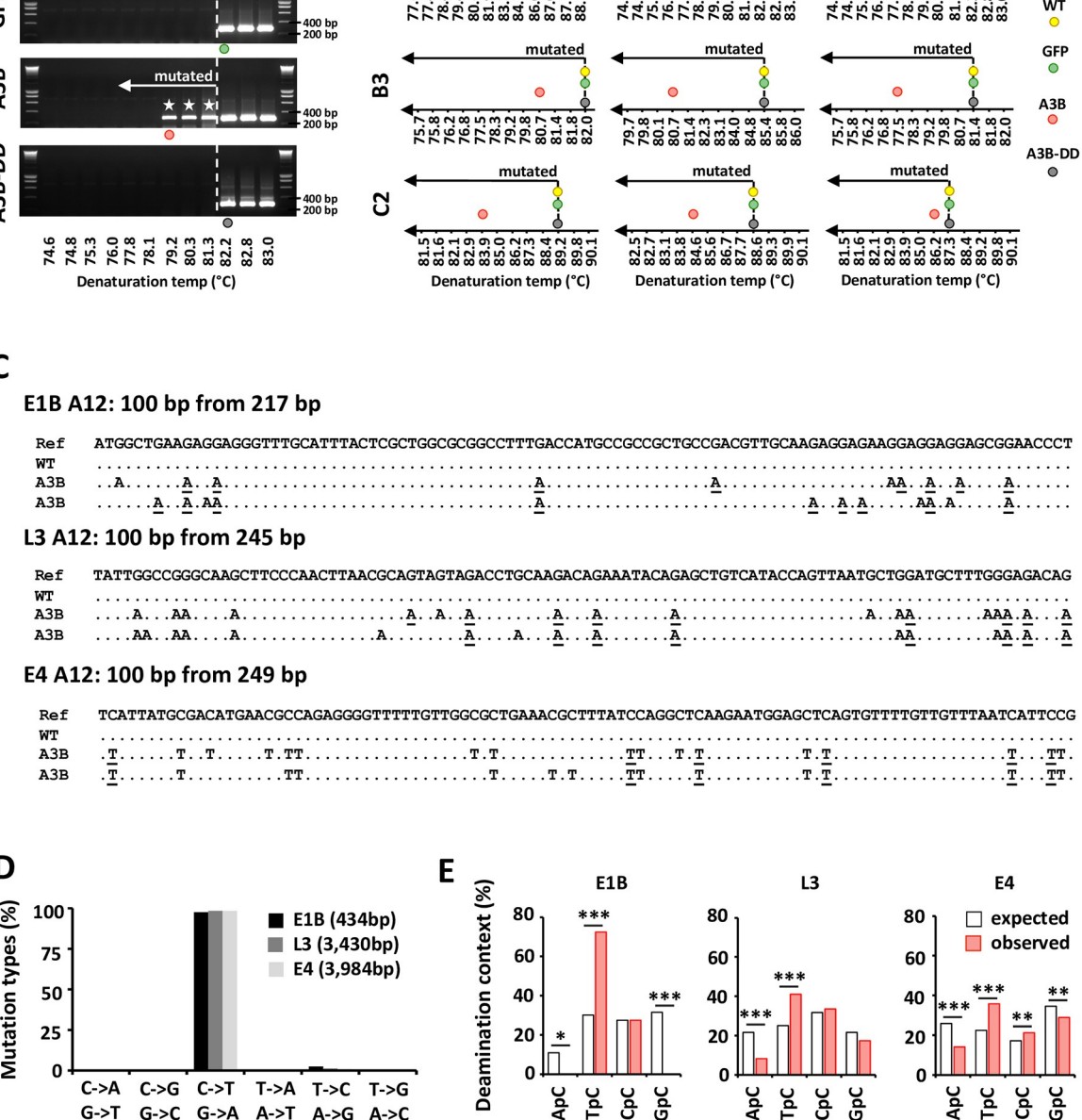

**Fig 4. A3B introduces strand-coordinated C to T clustered mutations in the adenovirus genomes.** HBEC-WT, -GFP, -A3B and -A3B-DD were infected with HAdV-A12, -B3 or -C2 and the intracellular DNA was extracted 48 hours post infection. 3DPCR reactions were conducted on the E1B, L3 and E4 genes. (A) Agarose gels illustrating 3DPCR reactions specific for the E4 gene of the A12 strain are shown. The dotted white line indicates the threshold between mutated and unmutated 3DPCR products. Low denaturation temperature amplicons were recovered in HBEC-A3B cells only. (B) Graphical representation of the 3DPCR results obtained for the 3 viral genes (E1B, L3 and E4), for the 3 viral strains (A12, B3 and C2) and for the 4 cell lines (WT, GFP, A3B and A3B-DD). The lowest denaturation temperatures allowing the production of the expected amplicons were represented on the gradients by colored circles. (C) 3DPCR products generated from HAdV-A12-infected WT and A3B-expressing cells were cloned and sequenced. Hypermutated sequences detected in A3B-expressing cells (A3B) were aligned against the reference viral genome (Ref) and against sequences isolated from WT infected cells (WT). Deamination occurred on the coding (C to T mutations) or the template strand (G to A mutations of the coding strand). The mutations underlined took place within an A3-favored motif with a T in 5' of the deaminated C. (D) Mutation types recorded on the E1B, L3 and E4 hypermutated sequences. The numbers in brackets indicate the number of bases sequenced. (E) 5' nucleotide contexts of the deaminated Cs recorded on the E1B, L3 and E4 hypermutated sequences were reported by red bars. 5' nucleotide context

expected values, based on the dinucleotide composition of the DNA sequences were represented by white bars. P-values were calculated by $\chi^2$-tests.

the E1B, L3 and E4 hypermutated sequences. This context is typical for the A3 proteins (with the sole exception of A3G that favors deamination of a C following another C).

Overall, these results demonstrate that A3B deaminates the viral genome of the three strains, on both strands and both at the ends and in the middle of the genome.

## A3B knockdown promotes replication of strain A12

We proved that exogenous expression of A3B restricts adenovirus replication. We next wondered whether knockdown of A3B could facilitate adenovirus replication. We used the A549 lung cancer cell line that constitutively expresses A3B (Fig 5B, first lane and S1 Fig for comparison with HBEC-WT cells). We transduced the A549 cells with lentiviral vectors expressing shRNAs against A3B (shA3B) or scramble control (scramble). Fig 5A demonstrates that A549-shA3B produces 26 times less mRNA that the WT or scramble control. Fig 5B shows that the A549-shA3B barely express the A3B protein and Fig 5C shows that A549-shA3B does not display deaminase activity, contrary to the WT or scramble controls.

Similarly with the previous experiment in HBEC3-KT cells, we infected the A549-WT, -scramble and -shA3B with the 3 HAdVs strains at a low MOI (0.03) and monitored the viral replication during several days post infection. Fig 5D shows the percentage of infected cells measured by flow cytometry during the first replicate. Data from the 3 replicates are summarized in Figs 5E and S5. We observed that the percentage of A12-infected cells is increased in the A549-shA3B cells compared to A549-WT cells at 4- and 7-days post infection (Fig 5E, left panel, blue bars vs orange bars), whereas no difference was observed for the B3 and C2 strains (Fig 5E, middle and right panels). Figs 5F and S6 show the intracellular viral load detected by qPCR at different time-points post-infection. We observed that the viral load in the A549-shA3B cells infected with the A12 strain (Fig 5F, left panel, blue bars) is increased at 2- and 4-days post infection. On the contrary, no difference was observed in the cells infected with the B3 or with the C2 strain (middle and right panels). Figs 5G and S7 show the amount of viral DNA in the supernatant. We observed that the viral load in the supernatant of the A549-shA3B cells infected with the A12 strain (Fig 5G, left panel) is increased at 7-days post infection. Once again, no difference was observed in the supernatant isolated from the cells infected with the B3 or with the C2 strain (middle and right panels).

To summarize, the down regulation of A3B speeds up the propagation of the A12 strain in cell culture, increasing its DNA replication and virion production. In other words, an endogenously-expressed A3B protein effectively restricts the propagation of the A12 strain. The replication of the two other strains, B3 and C2, is not impacted by the knockdown of the A3B gene.

## Hypermutated viruses are strongly reduced in A3B-knockdown cells

We then wondered whether the A3B knockdown would prevent the generation of C to U clustered mutations in the adenovirus genomes. Fig 6A illustrates the L3-specific 3DPCR amplicons obtained during HAdV-A12 infection of WT, scramble, and shA3B cells. We observed low denaturation temperature amplicons (Fig 6A) in the 3DPCR reactions performed on the DNA extracted from WT and scramble cells, but not from shA3B cells, indicating that A3B is necessary for the generation of hypermutated viruses. Fig 6B reports the 3DPCR results for the

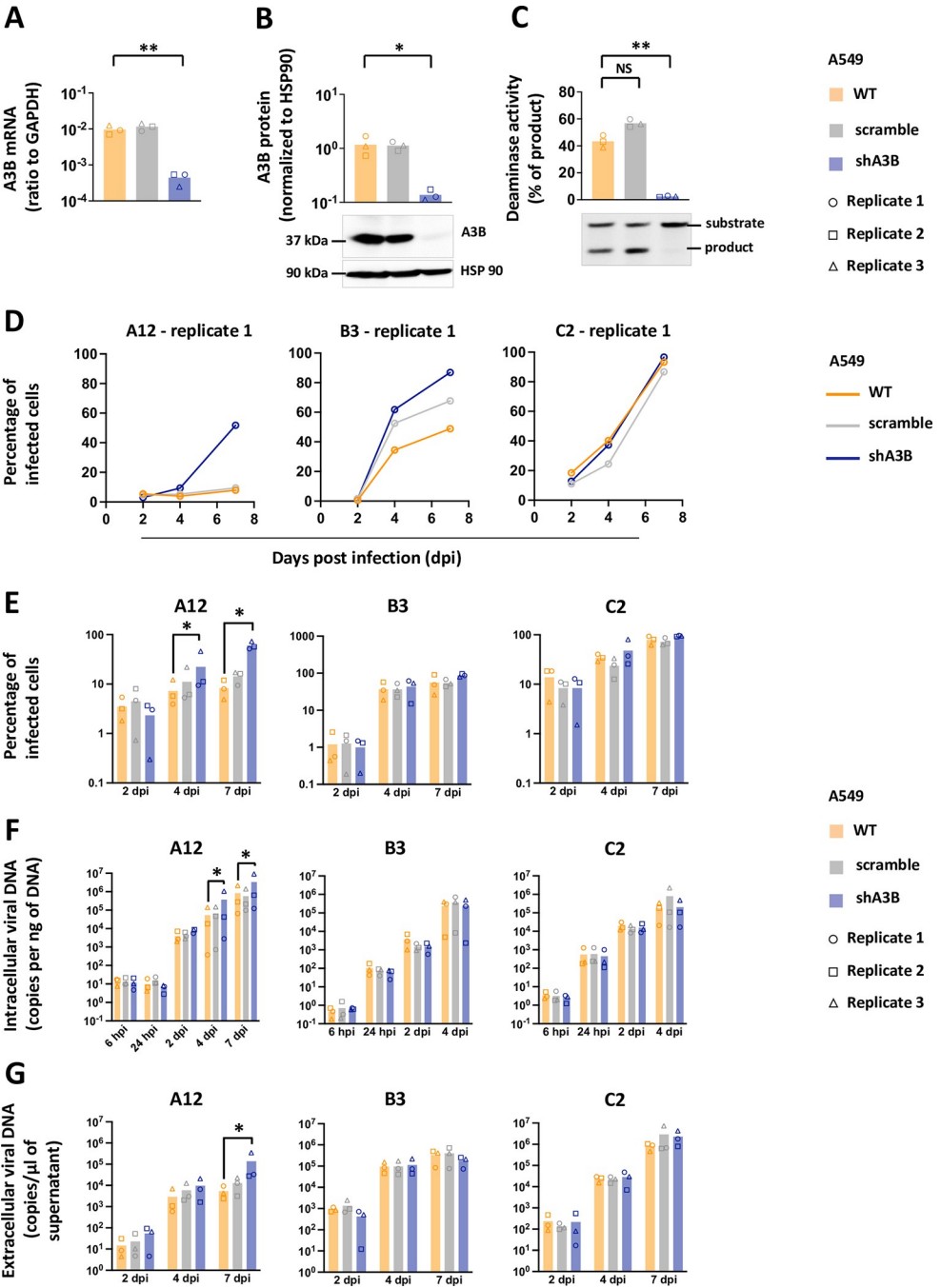

**Fig 5. A3B knockdown promotes replication of strain A12.** A549 lung cancer cells were transduced to stably express APOBEC3B-targeting shRNAs (shA3B) or a nontargeting control shRNA (scramble). (A) A3B mRNA levels were quantified by RTqPCR and expressed relative to GAPDH mRNA. (B) A3B protein levels were assessed by western blot, quantified by densitometry and normalized with HSP90. (C) Deamination activities of the cell lysates were expressed as the percentage of substrate converted into product and compared to the deamination activity of the WT cells. P-values were calculated by t-tests. (D-G) A549-WT, -scramble and -shA3B were infected with HAdV-A12, -B3 or -C2 at a MOI = 0.03 and analyzed at different time points post infection. (D-E) The percentage of infected cells were quantified by flow cytometry. The results of the first replicate are shown in panel D and the data from the 3 replicates are shown in panel E. Intracellular viral DNA levels (F) and extracellular viral DNA levels (G) were quantified by qPCR. P-values were calculated by ratio t-tests. Only significant differences are shown.

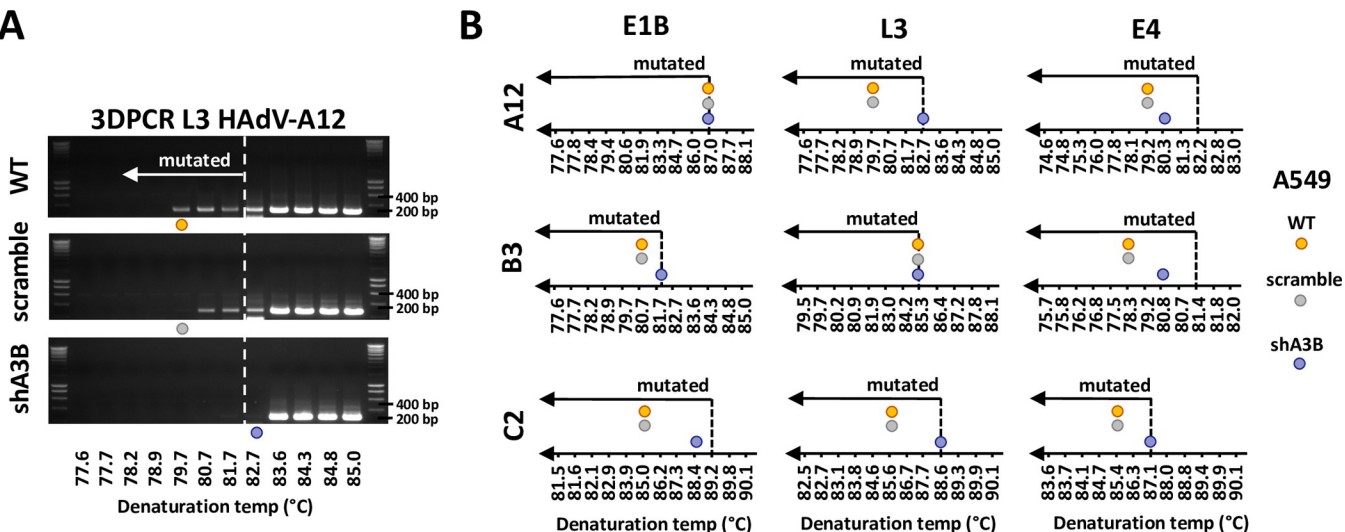

**Fig 6. A3B knockout reduces the presence of hypermutated viral sequences.** A549-WT, -scramble and -shA3B were infected with HAdV-A12, -B3 or -C2 and the intracellular DNA was extracted 48 hours post infection. 3DPCR reactions were conducted on the E1B, L3 and E4 genes. (A) Agarose gels illustrating 3DPCR specific for the L3 gene of the A12 strain. The dotted white line indicates the threshold between mutated and unmutated 3DPCR products. Low denaturation temperature amplicons were recovered in WT and scramble A549 cells. (B) Graphical representation of the 3DPCR results obtained for the 3 viral genes (E1B, L3 and E4), for the 3 viral strains (A12, B3 and C2) and for the 3 cell lines (WT, scramble and shA3B). The lowest denaturation temperatures allowing the production of the expected amplicons were represented on the gradients by colored circles.

3 viral genes (E1B, L3 and E4), the 3 viral strains (A12, B3 and C2) and the 3 cell lines (WT, scramble and shA3B). The dotted lines arbitrarily separating mutated from unmutated amplicons were placed based on the results obtained on the HBEC-WT cells which do not constitutively express A3B (Fig 1B). We observed that amplicons from WT and scramble cells generally display a lower denaturation temperature than the amplicons generated in A3B-knockdown cells.

This demonstrates that an endogenously expressed A3B can introduce mutation clusters in the adenovirus genome.

## Adenovirus infection leads to a reduction of the quantity and/or enzymatic activity of the A3B protein

We showed that A3B restricts the replication of adenoviruses with variable efficacies depending on the strain. The most sensitive strain being the A12, followed by the C2 and finally the B3 which is only marginally impacted by the A3B protein. We took advantage of the A549 cells which constitutively express A3B to test whether infection with the B3 and/or C2 strains can attenuate the A3B action. A549 cells were infected at a high MOI (MOI = 3) with each of the three strains. Fig 7A shows that infection of A549 cells with the A12 strain does not modify the level of the A3B protein, whereas a significant decrease of the protein amount can be detected in the cells infected with the B3 and the C2 strains after 1- and 2-days post infection (lanes 3, 4, 7 and 8). The transcription of the A3B gene can produce several isoforms. Thus, we first assessed by RT-PCR which isoform(s) is (are) produced. In the upper part of Fig 7B we depicted the four A3B mRNA isoforms (A3B-201 to 204). To determine which are the A3B isoforms expressed in uninfected control and in HAdV-infected cells, two regions of the A3B mRNA were amplified. The first region spans from exon 4 to exon 6; the second from exon 7 to exon 8. When amplifying the first region (using primers Ex4_F and Ex6_R), we only

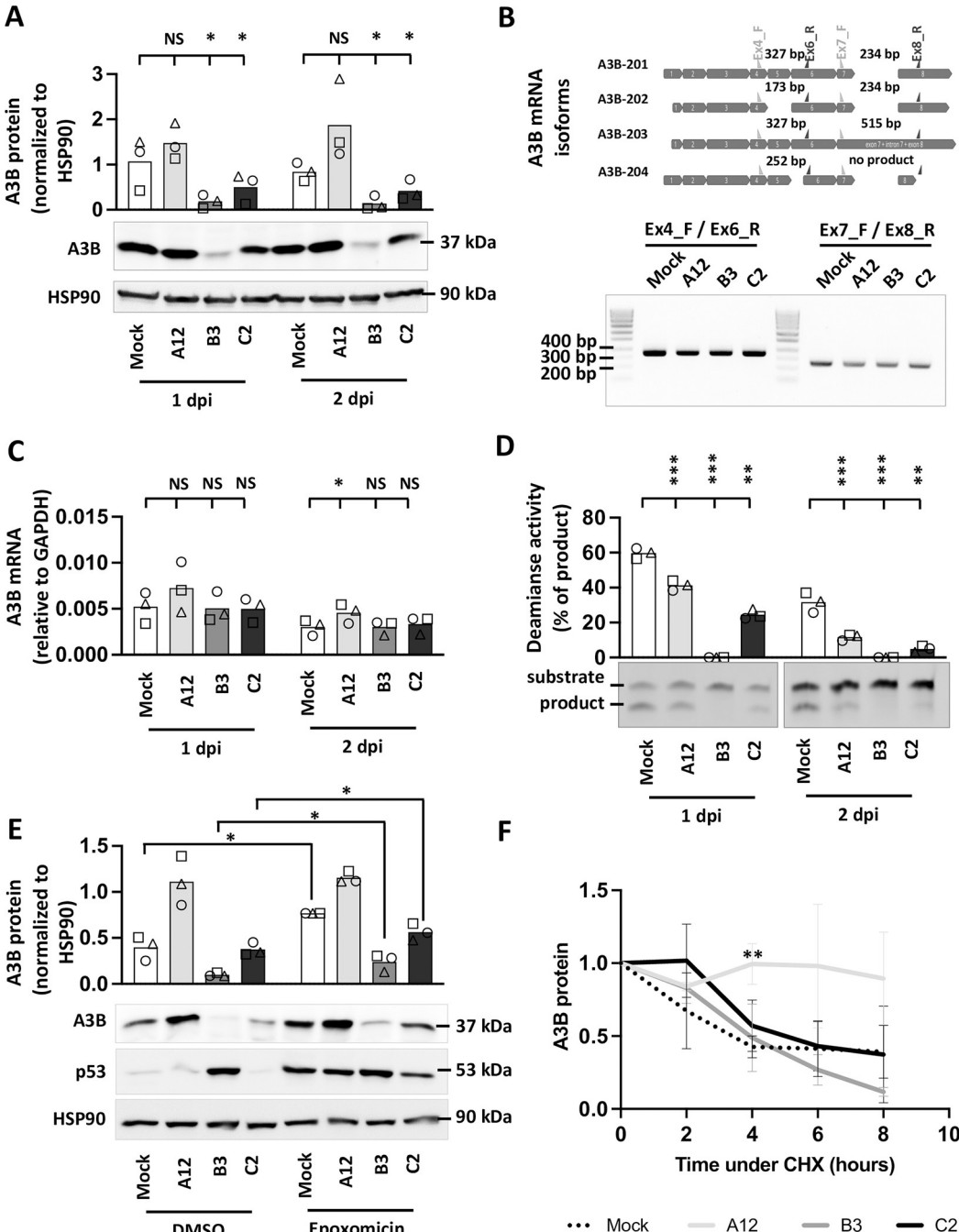

**Fig 7. Adenovirus infection reduces A3B quantity and/or enzymatic activity depending on the viral strain.** A549 cells were infected with HAdV-A12, -B3, -C2 or mock control at a MOI = 3. (A) A3B protein levels were assessed by western blot at 1- and 2-days post infection, quantified by densitometry and normalized with HSP90. (B) A3B mRNA isoforms were detected by discriminative RT-PCR reactions at 2-days post infection. (C) A3B mRNA levels were quantified by RTqPCR at 1- and 2-days post infection and expressed relative to GAPDH mRNA. (D) Deamination activities of the cell lysates were expressed as the percentage of substrate converted into product and compared to the deamination activities of the mock-infected cells at 1- and 2-days post infection. P-values were calculated by ratio t-tests. (E) Epoxomicin was added at 16 hpi. Six hours later, A3B, p53 and HPS90 protein levels were assessed by western blot. P-values were calculated by paired t-tests. (F) Cycloheximide was added at 16 hpi. The levels of A3B, p53 and HPS90 proteins were evaluated by western blot for a period of 8 hours and expressed relative to the level measured just before the addition of cycloheximide. P-values were calculated by t-tests (infected conditions compared to mock). Error bars show standard deviation.

detected a 327 bp-long PCR product both from control and infected cells (Fig 7B, left hand side of the gel). When using the second primers pair (Ex7_F and Ex8_R), we also detected a unique 234 bp-long PCR product both from control and infected cells (Fig 7B, right hand side of the gel). Taken together, these results showed no change in A3B isoform upon HAdV-infection. The A3B-201 mRNA isoform is the sole isoform expressed both in mock- and in HAdV-infected cells. Fig 7C shows that the levels of A3B mRNA were not impacted by the B3 or the C2 infection, and a slight increase can be observed in the A12-infected cells at 2-days post infection when compared with mock-infected cells. Therefore, the decrease of the A3B protein observed in the B3-, and to a lesser extent in the C2-infected cells, is not due to transcriptional changes. Fig 7D shows that the deaminase activity of the cell lysates is significantly decreased for the 3 strains, with the magnitude of the decrease being stronger in the B3- and C2-infected cells. The loss of deaminase activity observed in the cell lysates isolated from the B3- and C2-infected cells was expected, as the amount of the A3B protein has been significantly reduced. Importantly, we also observed that the deaminase activity in the A12-infected cells is being reduced (Fig 7D, lanes 2 and 6), although the infection with the A12 strain did not significantly modify the total amount of the A3B protein (Fig 7A, lanes 2 and 6). S8 Fig displays A3B protein levels versus deaminase activity of cell extracts for mock-, A12-, B3-, or C2-infected cells, after 1 and 2 dpi and for all 3 replicates. It illustrates that the deaminase activity in cells infected with A12 is strongly reduced despite a still high amount of A3B protein.

The decrease of the A3B protein amount observed in B3- and C2-infected cells could be due to an accelerated degradation of the A3B protein or to the inhibition of the translation of the A3B transcript. To better understand the mechanism involved, A549 cells were infected with the A12, B3 or C2 strain and epoxomicin, a proteasome inhibitor, was added to the culture medium at 16 hours post infection. A3B and p53 protein levels were assessed by western blot 6 hours after the addition of the proteasome inhibitor. Adenovirus species A and C species are known to promote p53 degradation in a proteasome-dependent way [26]. As expected, epoxomicin treatment inhibits the degradation of p53 by the A12 and C2 strains (Fig 7E). A3B protein levels are increased by epoxomicin treatment in mock-, B3- and C2-infected conditions (Fig 7E). These observations are consistent with work by Scholtés and colleagues identifying proteasomal activity as a post-translational mechanism regulating the amounts of A3 proteins [27]. We conclude that A3B degradation takes place both in mock-, B3- and C2-infected cells and that this degradation is dependent of the proteasome. It is important to note that epoxomicin treatment does not fully restore A3B level after B3 infection suggesting that it may have an additional mechanism leading to the decrease of A3B protein level. Interestingly, epoxomicin treatment did not impact the A3B protein level in A12-infected cells (Fig 7E), suggesting that A3B is not significantly degraded during the course of this experiment. Finally, A549 cells were infected with the A12, B3 or C2 strain and the translation inhibitor cycloheximide (CHX) was added to the culture medium at 16 hpi. A3B and p53 protein levels were assessed by western blot at 0, 2, 4, 6 and 8 hours after CHX addition (Figs 7F and S9 for representative western blot pictures). At 16 hpi, the level of the A3B protein is already largely decreased in B3- and C2-infected cells (S9 Fig). Fig 7F represents the A3B protein amount relative to the level measured at 16 hpi, just prior CHX addition. We observed that that A3B protein half-life is around 4 hours in mock-, B3- or C2-infected cells. On the contrary, the infection with the A12 strain seems to stabilize the A3B protein (reaching statistical difference at 4 hours post CHX treatment compared to mock-infected cells).

We conclude that B3 and C3 adenovirus infections lead to a reduction in the A3B protein, which reduction depends on proteasomal activity. Infection with the A12 strain is accompanied by stabilization of the A3B protein and a decrease in its enzymatic activity.

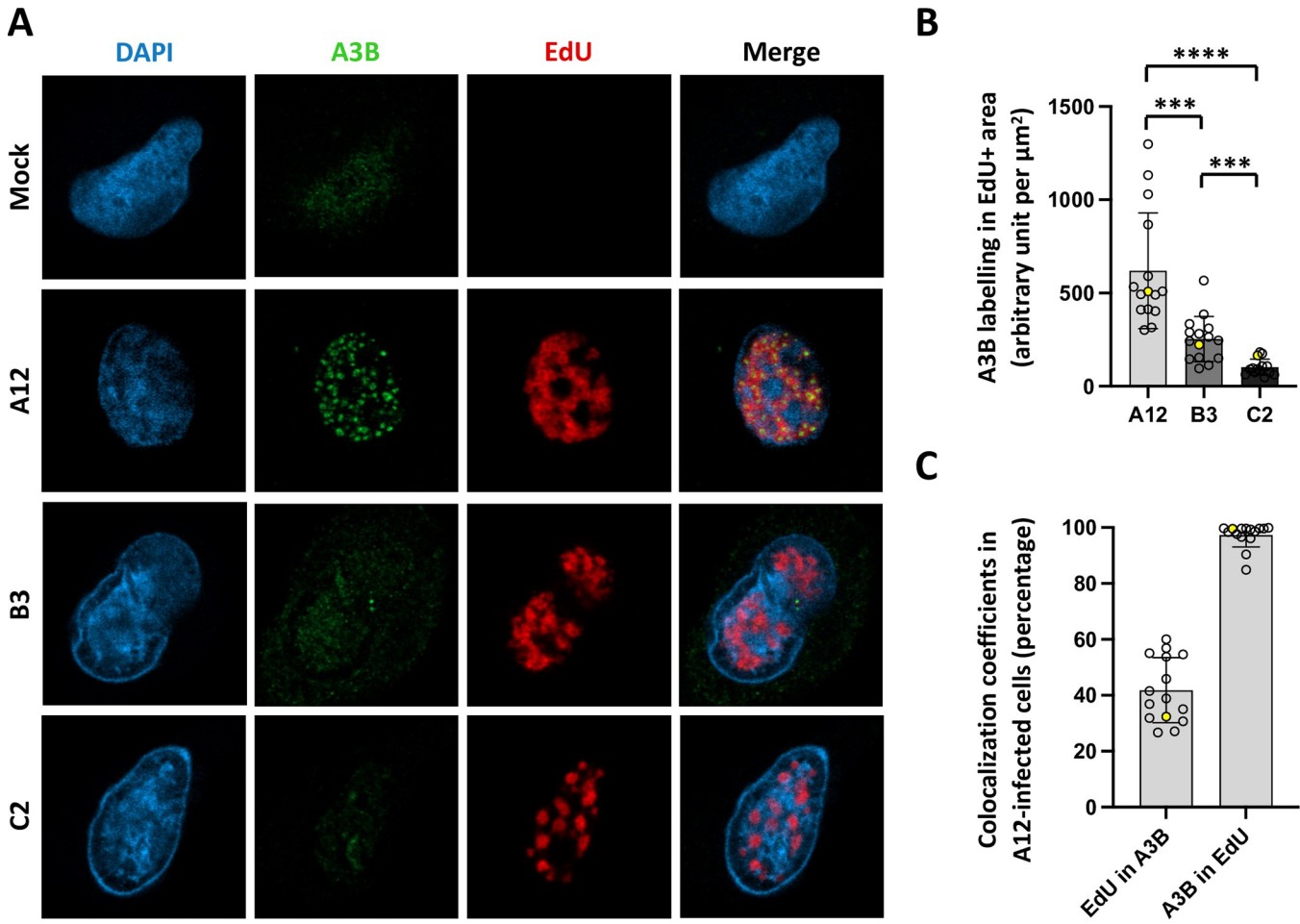

**Fig 8. A3B recolocalizes inside the viral replication centers in A12-infected cells.** A549 cells were infected with HAdV-A12, -B3, -C2 or mock control at a MOI = 3. B3- and C2-infected cells were analyzed 20 hours post infection and A12-infected cells were analyzed 24 hours post infection. (A) The A3B protein (green) and the viral replication centers as identified by EdU labelling (red) were imaged by fluorescence microscopy. (B) A3B labelling was quantified within the EdU+ area. Each circle represents a given cell. The yellow-colored circles correspond to the images shown in panel A. P-values were calculated by unpaired t-tests. (C) Colocalization between A3B and EdU labeling in A12-infected cells was estimated using Manders' coefficients.

## A3B colocalizes with the viral replication centers in A12-infected cells

Exclusion of the A3B protein from the viral replication centers is one of the mechanisms used by herpesviruses to evade the A3B-mediated response [14]. Fig 8A illustrates the subcellular distribution of A3B (in green) during the infection of A549 cells by the adenoviruses A12, B3 and C2. The viral replication centers are labelled by the incorporation of EdU and colored in red. The A3B protein is distributed homogeneously in the nucleus of mock-infected cells. The A3B labelling is weak in C2- and B3-infected cells. On the contrary, the A3B protein forms dense clusters in the nucleus of A12-infected cells (Figs 8A and S10A for additional pictures). As labelling control, A549 shA3B cells were infected with the A12 strain. No A3B clusters could be observed in those cells (S10 Fig). Fig 8B shows that A3B labelling is stronger in the viral replication centers from A12-infected cells than those of B3- or C2-infected cells. Fig 8C show that, on average, 97% of the A3B labeling in A12-infected cells is localized inside the EdU-positive areas.

We conclude that, in A12-infected cells, the A3B protein relocalizes inside the viral replication centers.

## Adenoviruses of the species A display a significant APOBEC3 evolutionary footprint

In this study, we have shown that A3B restricts adenoviruses with varying efficacies between strains, the B3 and C2 being less impacted by A3B than A12. We wondered whether the ability to resist A3B is shared between strains of a given species. We choose to tackle that question by looking at the genomic sequences of adenoviruses. We have previously reported that the A12 strain bears a stronger APOBEC3 footprint than the B3 and C2 strains [22]. We now wondered whether the A12 was the sole strain showing such footprint or whether it is shared by other adenoviruses. As already explained, A3 proteins favor deamination of C when preceded by a T. Thus, the A3s will turn 5'TC motifs into 5'TU dinucleotides in the genome of exposed viruses. Depending on the position of the mutated C within the codon, the mutation can be synonymous or nonsynonymous. When the mutated C is at the third position of the codon, deamination of the C will turn the NTC codon into an NTT codon. This mutation will always be synonymous (symbolized by an "Syn" in Fig 9A). When A3-related deamination intervenes on a C located at the first position of a codon, the NNTCNN motif will be converted into an NNTUNN motif and will produce a nonsynonymous mutation (symbolized by an "NSyn" in Fig 9A). Similarly, deamination of a C located at the second position of the codon will convert a TCN codon into a TUN codon and most likely introduce a nonsynonymous mutation. Because a synonymous mutation is more likely to be conserved than a nonsynonymous, the A3-driven natural selection should deplete more intensively NTC codons than TCN or NNTCNN motifs (as in those cases the C to U mutation will impact the encoded amino acid). Thus, we defined the APOBEC3 footprint as an under-representation of NTC codons because A3s favor 5'-TC motifs and because a C to U mutation in the third position of a codon is likely to be retained. Fig 9B shows that adenoviruses of the species A display an occurrence of NTC codons lower than expected. The level of NTC under-representation is similar between the four strains of adenoviruses A (namely A12, A18, A31 and A61) and is lower than any other species. We speculate that, like the A12, the other A strains are less able to antagonize A3B and therefore evolved to reduce the number of A3-favored motifs. Within the different species, the A3 footprint is relatively homogenous. The B, D, E and G species show no NTC depletion on average for the whole viral genome, whereas the C and the F species display a moderate under-representation of NTC codon.

These data corroborate our *in vitro* observations and suggest that the capacities to antagonize A3B are shared among members of a given species.

## Discussion

### How frequent are the (hyper)mutated sequences?

In our tests, the three adenoviral strains were not equally sensitive to A3B, the replication kinetics of the B3 and C2 were much less impacted by A3B than that of A12. We did retrieve hypermutated viral sequences for the three strains, sequences that most likely will not lead to infectious viral particles. However, the 3DPCR does not allow their quantification. Thus, we speculate that the proportion of viruses carrying mutation clusters is higher during infection with the A12 compared to the other two strains. A formal investigation will require the set-up of a new technique based on high throughput sequencing. The quantification of the proportion of viruses carrying C to T clustered mutations might be important for the field of adenovirus-based vectors and vaccines. If hypermutated sequences are generated during vectors production, it will be interesting to measure their frequency and their impact on production yield. The knockdown of the A3B gene in the helper cell lines might improve adenoviral vector production and quality.

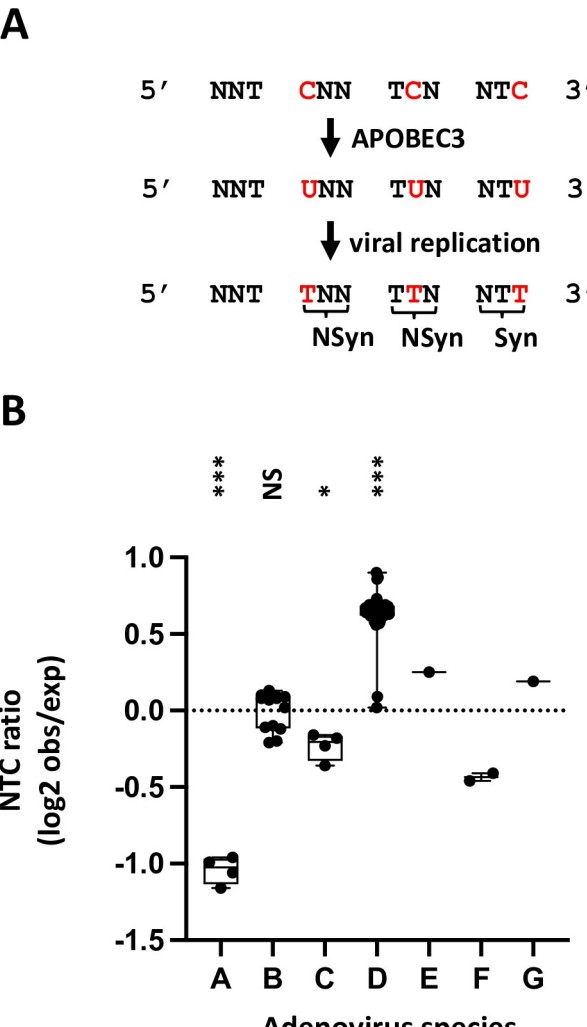

**Fig 9. Adenoviruses of the species A display a significant APOBEC3 evolutionary footprint.** (A) The A3s induce C-to-U deamination preferentially in a 5'TC dinucleotide context. After viral replication the mutation is conserved in the viral genome as a C to T transition (in red). Depending on the position of the TC motif in codon, the mutation is synonymous (Syn) or non-synonymous (NSyn). Since synonymous mutations are more likely to be conserved, the APOBEC3 footprint is defined as the depletion of the NTC codon. (B) The NTC codon observed/expected ratios were calculated in 70 different adenoviruses spanning the 7 species. P-values were calculated by t-tests.

## How A3B is being recruited in the viral replication centers?

We observed that A3B restricts adenoviruses by a deaminase-dependent mechanism and showed that it colocalizes with the virus replication centers (as identified by their high level of EdU incorporation in A12-infected cells). This suggests that A3B edits adenoviruses during viral DNA replication. Of note, adenoviruses replicate by strand displacement. The displaced strand, being single-stranded, represents the prototypical substrate for A3B editing. It will be interesting to test whether A3B can be actively recruited to the displaced strand and whether it can replace some of the viral DNA-binding proteins (DBP) to act on the viral DNA.

Recruitment of A3B to virus replication centers in A12-infected cells is reminiscent of observations made in BK PyV-infected cells where A3B appears to access at least a subset of BK PyV replication centers [16]. On the other hand, it contrasts sharply with the active

exclusion of A3B from replication centers of herpes viruses. Indeed, EBV and other herpes viruses like KSHV and HSV-1 encode viral proteins that relocate A3B to perinuclear bodies [13,14]. How adenovirus A12 and BK PyV cope with the presence of A3B in their genome replication centers remains an open question but low A3B editing activity might be beneficial for the genetic diversification of the virus (as discussed below).

## How does the A3B-mediated response against adenovirus differ between HBEC3-KT and A549 cells?

We recently reported that, upon adenovirus infection of HBEC3-KT cells, the A3B mRNA levels were increased [22]. In this work, we observed that infection of A549 cells with the A12 strain triggers a modest increase of A3B mRNA, but no change can be observed in A549 infected with the B3 or C2 strain (Fig 7C). Importantly, the basal levels of A3B (mRNA and protein) are different between HBEC3-KT and A549. HBEC3-KT are normal immortalized human bronchial epithelial cells and they do not constitutively express A3B (Figs 1B and S1). On the contrary, A549 cells are lung cancer cells that constitutively express an enzymatically active A3B protein (Figs 5B and S1). Since A3B is already strongly expressed in A549 cells, adenoviral infection seems to only marginally increase its transcription in these cells. At the protein level and in A549, A3B was not changed upon infection with the A12 strain whereas it decreases upon B3 and C2 infection (Fig 7A). In HBEC3-KT cells, the A3B protein was strongly increased upon infection with the A12 strain, slightly increased with the C2 and no difference could be evidence after infection with the B3 strain [22]. The same processes might be in place, where the discrepancy between mRNA and protein levels in B3 and C2-infected cells can be explained by an antagonization of A3B.

## Do adenoviruses actively antagonize A3B, by which mechanisms?

Our study shows that adenoviruses antagonize A3B through the decrease of its quantity and/ or enzymatic activity, depending on the strain. Further experiments will be necessary to pinpoint the mechanisms and the viral proteins involved. We observed that the A3B protein half-life in B3- or C2-infected cells is not different from its half-life observed in mock-infected cells. This suggests that the decrease of the A3B amount observed in the B3- and C2-infected cells is rather due to an inhibition of the translation of A3B mRNA. On the contrary, in A12-infected cells, our data point toward a stabilization of the A3B protein and an inhibition of its enzymatic activity. It has been shown that A3B protein can be inactivated by the phosphorylation of its threonine 214 by the cellular kinase PKA [28]. Interestingly, the E1A protein is known to mimic a cellular A-kinase anchoring protein (AKAP) and relocates PKA to the nucleus [29]. It would be interesting to assess the phosphorylation profile of A3B in A12-infected cells and test the potential involvement of E1A.

Understand the mechanisms in place to downregulate/inactivate A3B might have implications for anticancer treatment and for the design of oncolytic adenoviruses. Indeed, upregulation of A3B has been observed in many tumors types and has been demonstrated to contribute to the A3 mutational signature [30–32]. Moreover, A3B expression in tumor has been shown to fuel genetic diversity of the tumor cells, favoring metastasis and drug resistance [33–35]. Finally, one of the limitations of oncolytic viruses' efficacy, is the expression by the host cells of the A3B protein [36]. Hence, oncolytic adenoviruses particularly efficient at antagonizing A3B might be more able to propagate in the tumor and therefore at killing it. Also, the use of adenoviruses or specific adenovirus proteins to deplete the tumor from its A3B protein might improve efficacy of anticancer treatments and relapse-free survival.

### Can adenoviruses benefit from a low A3B activity?

We previously reported that adenovirus genomic sequences display a specific codon bias (i.e. depletion of NTC codons) suggesting evolution under the selection pressure of A3s [21]. In this work we report that A3B can introduce strand-coordinated C to T clustered mutations the adenovirus genome, leaving those viruses indubitably noninfectious. If hypermutated sequences are easy to identify, they likely represent only a small fraction of the edited sequences, the vast majority of sequences being slightly to not edited at all. Once again, a comparison with other viruses might be informative. Thus, HIV Vif antagonization of A3G is not 100% effective and some low A3G editing remains, fueling genetic diversity and emergence of drug-resistant variants [37]. Similarly, mutations in the BK PyV capsid accumulate in kidney transplant recipients with persistent virus replication. These mutations confer resistance to neutralizing antibodies and are likely to be the result of an A3 activity [19]. By analogy, it would be interesting to test whether A3B could also be diverted by the adenoviruses to fuel antigenic drift or drug evasion. We observed that A3B restricts *in vitro* the A12 strain more efficiently than the B3 and C2. One might wonder whether the A12 actually requires more editing than other strains to evade the immune response.

### Do adenoviruses play a role in the deregulated expression of A3B in human tumors?

Even though the A3 enzymes are potent effectors protecting the host against viral infections, the A3s can also accidently mutate the cellular genome. Mutations attributed to A3 activity are found in many cancer types such as bladder, bone, cervical, breast, lung, and head and neck cancer [30,31,38–40]. These mutations appear to be mostly introduced by the A3A and A3B proteins [41,42]. The deregulated expression of A3A and A3B in tumor appears to be the result of viral infection and/or dysregulated inflammatory response. For instance, HPV-16 and -18 infections cause cervical cancer and head and neck squamous cell carcinoma, tumors that bear a high level of A3-induced mutations, resulting from the upregulation of both A3A and A3B via the oncoproteins E6 and E7 [39,43]. Involvement of PyVs in bladder cancer has long been suspected and the A3 mutational signature found in bladder tumors has been proposed to be the result of a hit-and-run mechanism where PyV triggers a strong A3B expression [44]. It must be stated clearly that the present study does not bring evidence that adenoviruses cause cancer in Human. However, by analogy with the discoveries made on HPVs and PyVs and because HAdVs can also persistently infect some specific cell types, we consider that studies looking for the involvement of adenoviruses in human cancer should be pursued.

To summarize, this work identifies A3B as an adenovirus restriction factor, limiting virus replication by introducing mutations in the viral genome. Depending on the viral strains, A3B restricts viral replication with different efficiency; the strains less sensitive having the ability to antagonize the A3B protein by decreasing its amount. In a larger picture, it would be interesting to investigate the impact of the deletion of the A3B gene in helper cell lines used to produce adenoviral vectors. The precise characterization of the viral proteins involved in the A3B downregulation/inactivation may also have implications for the design of oncolytic adenoviruses, as A3B expression by the cancer cells has been shown to limit the efficiency of oncolytic viruses [36]. Finally, the impact of adenoviral infections on A3B upregulation deserves to be investigated in detail, in the context of A3-driven tumors.

## Material and methods

### Cell lines and drugs

HBEC3-KT cells are normal human bronchial epithelial cells immortalized with the human *TERT* and mouse *Cdk4* genes. HBEC3-KT cells were kindly provided by Prof. Jerry W. Shay

(UT Southwestern, Dallas, Texas, USA) [45]. HBEC3-KT were cultured in keratinocyte-SFM medium supplemented with bovine pituitary extract (50 μg/mL) and human recombinant epidermal growth factor (5 ng/mL) (Gibco) on gelatin-coated flasks. HBEC3-KT constitutively expressing GFP (HBEC-GFP), A3B (HBEC-A3B) or A3B Deaminase Dead (HBEC-A3B-DD) were established by transduction with lentiviral vectors encoding for a GFP, A3B or A3B-DD protein (pLenti4-A3B and pLenti4-A3B-E68Q-E255Q encoding respectively the catalytically active and inactive A3B were kindly provided by Prof. Reuben Harris) [46]. The human lung carcinoma cell line A549 (American Type Culture Collection) and PKR-deficient A549 cell line were cultured in Dulbecco's Modified Eagle's medium (Gibco) supplemented with 10% fetal calf serum (Gibco) and 10 mM of L-glutamine. PKR-deficient A549 cells were kindly provided by Dr. Cheng Huang and Prof. Slobodan Paessler (UTMB, Galveston, Texas, USA) [47]. A549 cells knock-down for A3B (A549 A3B shRNAs) were established by transduction with lentiviral vectors encoding A3B-targeting shRNAs (pSicoR-MS2 plasmids, kindly provided by Warner C. Greene, University of California, San Francisco, CA, USA) [12]. A549 cells were also transduced with a lentiviral vector encoding non-targeting shRNA (A549 NTC shRNA) and used as control [12]. Of note, the puromycin markers originally present in the pSicoR-M2 and pLenti4-A3B plasmids have been replaced by a blasticidin resistance gene prior to A549 or HBEC3-KT transduction. Blasticidin was used to select for a polyclonal population of effectively transduced cells. All cell lines were incubated at 37°C and 5% $CO_2$. Epoxomicin (Calbiochem) was used at a concentration of 5 μM. Cycloheximide (Sigma Aldrich) was used at a concentration of 100 μg/ml.

## Virus preparation and infection procedure

Handling of human adenoviruses was done in a biosafety level 2 laboratory. Adenovirus A12 (HAdV-12, ATCC VR-863) was purchased at the American Type Culture Collection. Adenoviruses B3 and -C2 were kindly provided by Prof. Lieve Naesens (KULeuven, Leuven, Belgium) and strain identity was verified by genotyping as published in [22] and using primers described in [48]. HAdV-B3 and -C2 viral stocks were produced in A549 cells and HAdV-A12 viral stock in PKR-deficient A549 cells. During viral stock production, A549 cells and PKR-deficient A549 cells were cultured in OPTI-MEM Reduced-Serum Medium without phenol red (ThermoFisher Scientific). When 80% of the cells showed cytopathic effects, cells were scraped, collected with the culture medium and centrifuged at 3,500g for 15 minutes. The supernatant was collected and set aside. The cell pellet was resuspended in 2 mL of culture medium, submitted to 3 freeze/thaw cycles and centrifuged at 3,500g for 15 minutes. The supernatant was collected, pooled with the previous supernatant and treated with the endonuclease Benzonase (1 U/mL) (Sigma-Aldrich) at 37°C for 30 minutes to degrade the unpackaged nucleic acids. Supernatant was then filtered on a 0.22 μm Steriflip Filters (Merck Millipore) to remove residual cell debris. Eluate was collected and further filtered through Amicon Ultra-15 Centrifugal Filter Unit 100 KDa (Merck Millipore). Virions were retained on the filter, resuspended in PBS and stored at -80°C. Titration of the viral stocks were done by fluorescent forming unit (FFU) assay and by qPCR quantification of genome copies (details below). HBEC3-KT cells were infected with the HAdV-A12, -B3 or -C2 strain at a multiplicity of infection (MOI) of 0.03 or 3 in keratinocyte-SFM medium supplemented with bovine pituitary extract (50 μg/mL) and human recombinant epidermal growth factor (5 ng/mL) (Gibco) on gelatin-coated flasks. A549 cells were infected with the HAdV-A12, -B3 or -C2 strain at a multiplicity of infection (MOI) of 0.03 or 3 in Dulbecco's Modified Eagle's medium (Gibco) supplemented with 2% fetal calf serum (Gibco) and 10 mM of L-glutamine.

## Viral and cellular mRNA quantification by RT-qPCR

At the indicated time post infection, cells were collected, washed in PBS and resuspended in 1 mL of TRIzol reagent (Invitrogen) for RNA extraction. Two hundred μL of chloroform were added and samples were centrifugated at 4˚C at 12 000g for 15 minutes. Five hundred μL from the aqueous phase was collected and added to 450 μL of isopropanol. Samples were centrifuged at 12,000g for 15 minutes. Pellets were then washed in 900 μL of 75% ethanol, centrifuged at 12,000g for 10 minutes and resuspended in 30 μL of RNAse free water. cDNA was obtained by reverse transcription of 1 μg of RNA using iScript cDNA Synthesis Kit (Bio-rad) following manufacturer's instructions. The cDNA was subjected to qPCR using Takyon No Rox SYBR 2X MasterMix blue dTTP (Eurogentec) and CFX96 1000Touch Real-Time PCR System (Biorad). The primers used to amplify the A3B cDNA were designed by [49]. The A3B, E1A, E4orf6 and penton levels were expressed relatively to the abundance of GAPDH. Primers are listed in Table A in S1 Text.

## Viral copy number quantification by qPCR

DNA was extracted from infected cells using Nucleospin Tissue (Macherey Nagels) according to manufacturer's instructions and subjected to PCR. In parallel, supernatant containing virions was inactivated 15 minutes at 70˚C and viral copies were directly quantified. A fragment of the penton gene was amplified using Takyon No Rox SYBR 2X MasterMix blue DTTP (Eurogentec) and CFX96 1000Touch Real-Time PCR System (Biorad). Serial dilutions of plasmids containing the target sequences were used as calibration curves. Primers are listed in Table B in S1 Text.

## Infectious viral particles quantification by foci forming assay

A549 cells were infected with serial dilutions of HAdV stocks or supernatants containing virions in DMEM supplemented with 2% FCS. After 24 hours, cells were fixed with a mix of methanol/acetone 1:1 and incubated 5 minutes at -20˚C. Cells were then washed once with PBS and permeabilized using the permeabilization buffer (Invitrogen) according to the manufacturer's instructions. Cells were then incubated 1 hour at 4˚C with anti-HAdV antibody (MAB 8051, Sigma-Aldrich) used at a dilution of 1:200 in permeabilization buffer supplemented with 2% of goat serum (Gibco). After 2 washes in permeabilization buffer supplemented with goat serum, cells were incubated 1 hour with an anti-mouse AlexaFluor 568 (Invitrogen). Cells were washed twice with permeabilization buffer supplemented with 2% of goat serum and once in PBS. Finally, slides were mounted in Fluoromount G (Invitrogen) and imaged using Leica SP5 microscope. Infected cells were counted to calculate the number of foci forming unit (FFU) and expressed per μL of the starting infectious material.

## Percentage of infected cells by flow cytometry

Two-, four- or seven-days post infection, cells were collected and washed in PBS. Cells were fixed and permeabilized using the eBiosciences Foxp3/Transcription Factor Staining Buffer set (Invitrogen) according to manufacturer's instructions. Cells were then incubated 30 minutes at 4˚C with anti-HAdV antibody (MAB 8051, Merck Millipore) used at a dilution of 1:200 in permeabilization buffer supplemented with 2% of goat serum (Gibco). After 2 washes in permeabilization buffer supplemented with goat serum, cells were incubated 30 minutes with an anti-mouse antibody coupled with phycoerythrin (Miltenyi Biotec). Cells were washed two times with permeabilization buffer supplemented with 2% of goat serum and then resuspended

in PBS. Percentage of HAdV-infected cells was quantified using BD FACSVerse Flow Cytometer and BD FACSuite v1.0.6 software (BD Biosciences).

## A3B protein detection by immunoblotting

Cells were collected, washed in PBS and resuspended in HED buffer (20mM HEPES pH 7.4, 5mM EDTA, 1mM DTT, 10% glycerol) supplemented with cOmplete Protease Inhibitor Cocktail (Roche). Cells were then submitted to one freeze/thaw cycle and sonicated 15 cycles of 30 sec ON / 30 sec OFF using Bioruptor Pico device (Diagenode) at 4°C. Cell lysates were spun down at 14,000 rpm for 15 minutes to remove cell debris. Proteins were quantified using Pierce BCA Protein Assay Kit (ThermoFisher Scientific). Twenty-five μg of proteins were loaded on a 10% SDS-PAGE gel and transferred actively to PVDF membrane (GE healthcare Life Sciences). Membrane was blocked in TBS (Tris Buffered Saline) supplemented with 0.1% Tween 20 and 5% bovine serum albumin (BSA). A3B protein was detected with anti-Human APOBEC3B monoclonal antibody (5210-87-13, cat# 12397, from Prof. Reuben Harris and obtained through the NIH AIDS Reagent Program, Division of AIDS, NIAID, NIH [50]) used at a dilution of 1:1000 in TBS supplemented with 0.1% Tween 20 and 5% BSA. Hsp90 was used as loading control and was detected with an anti-HSP90AB1 antibody (Sigma-Aldrich) at a dilution of 1:1000 in TBS supplemented with 0.1% Tween 20 and 5% BSA. An HRP-coupled anti-rabbit IgG secondary antibody (Dako) was used at a dilution of 1:2000 in TBS with 0.1% Tween 20 and 5% BSA or 4% milk. Membranes were incubated with the chemoluminescent SuperSignal West Femto Maximun Sensitivity Substrate (ThermoFisher Scientific) and chemoluminescence was read using an ImageQuant LAS4000 (GE Healthare Life Sciences). Densitometry analysis was done using ImageJ software [51].

## Deamination assay

One or two-days post infection, cells were collected and washed in PBS. Cells were resuspended in HED buffer (20mM HEPES pH 7.4, 5mM EDTA, 1mM DTT, 10% glycerol) supplemented with cOmplete Protease Inhibitor Cocktail (Roche). Cells were then submitted to one freeze/thaw cycle and sonicated 15 cycles of 30 sec ON / 30 sec OFF using Bioruptor Pico device (Diagenode) at 4°C. Cell lysates were spun down at 14,000 rpm for 15 minutes to remove cell debris. Proteins were quantified using Pierce BCA Protein Assay Kit (ThermoFisher Scientific). Forty μg of proteins were incubated overnight at 37°C with 1 pmol of a fluorescent oligo substrate (5'-ATTAT TATTATTCAAATGGATTTATTTATTTATTTATTTATTT-Cy5-3'), 1mM $ZnCl_2$, 0.025U uracil DNA glycosylase (NEB), 2 μL 10x UDG buffer (NEB) and 100 μg/mL RNAse A (Thermofisher Scientific). Reaction mixture was treated with 50 mM NaOH and heated to 95°C for 10 minutes to cleave DNA probes at the abasic site. Reaction mixture was then neutralized with 50 mM HCl and mixed with 1.25x formamide buffer. Substrates (43 bases-long) were separated from products (30 bases-long) on a 15% tris-borate-EDTA (TBE)-urea gel. The Cy5-labeled substrates and deamination products were detected using ImageQuant LAS4000 mini (GE Healthcare Life Sciences). Densitometry analysis was done using ImageJ software [51].

## Selective amplification of hypermutated viral sequences by 3DPCR

Total DNA from infected cells was extracted using the MasterPure complete DNA and RNA purification kit (Epicentre) and resuspended in 50 μL sterile water. All amplifications were performed using first-round standard PCR followed by nested 3DPCR [52,53]. PCR was performed with 1U Taq DNA polymerase (Bioline) per reaction. After purification, PCR products were cloned into TOPO 2.1 vector (Life Technologies) and sequencing was outsourced to Eurofins Genomics. Primers are listed in Table C in S1 Text.

## Discrimination of the different A3B transcriptional isoforms by RT-PCR

Two-days post infection, cells were collected, washed in PBS and resuspended in 1 mL of TRIzol reagent (Invitrogen) for RNA extraction as described above. Samples were then treated with the TURBO DNA-free kit (Invitrogen) to remove DNA contaminants. cDNA was obtained by reverse transcription of 1 μg of mRNA using iScript cDNA Synthesis Kit (Bio-rad) following manufacturer's instructions. cDNA was subjected to PCR using the Go Taq G2 polymerase (Promega) using two sets of primers enabling discrimination of the four A3B transcriptional isoforms. Primers are listed in Table D in S1 Text.

## A3B and viral replication centers subcellular localization by immunofluorescence microscopy

Nineteen- or twenty-three-hours post-infection, A549 cells were treated with EdU (Click-iT Plus EdU Cell Proliferation Kit for Imaging, Invitrogen) at a final concentration of 10 μM according to the manufacturer's instructions. After one hour of incubation, cells were fixed with a mix of methanol/acetone 1:1 and incubated 5 minutes at -20˚C. Cells were then washed once with PBS and permeabilized using the permeabilization buffer (Invitrogen) according to manufacturer's instructions. Cells were then incubated overnight at 4˚C with anti-Human APOBEC3B monoclonal antibody (5210-87-13, cat# 12397) used at a dilution of 1:250 in permeabilization buffer supplemented with 2% of goat serum (Gibco). After 2 washes in permeabilization buffer supplemented with goat serum, cells were incubated 1 hour with an anti-rabbit AlexaFluor 488 (Invitrogen). Cells were washed twice with permeabilization buffer supplemented with 2% of goat serum. EdU labelling was then performed according to manufacturer's instructions. Cells were washed twice with permeabilization buffer supplemented with 2% of goat serum and once in PBS. Finally, slides were mounted in Fluoromount G (Invitrogen) and imaged using Zeiss LSM 900 Airyscan 2 Multiplex microscope. Images were then processed using Fiji software [54] and colocalization coefficients were calculated using Manders' method [55].

## APOBEC3 evolutionary footprint by bioinformatic analysis

Complete HAdV genomes were downloaded from "NCBI Nucleotides" database. Calculation of the NTC ratio was done as described in [21]. Briefly, the NTC ratio is given as the log2 ratio of the observed occurrence of the NTC codon to its expected occurrence. The NTC codon includes the ATC, CTC, GTC and TTC codons. To calculate the expected occurrence of the NTC codon, each coding sequence has been randomly shuffled a thousand times, retaining only the nucleotide composition. The expected count of NTC codons is calculated as the average of the occurrences of this codon over the thousand iterations. To calculate the NTC ratio for an entire viral genome, a "synthetic coding genome" was generated by concatenating the different coding sequences. The synthetic coding sequence is then randomly shuffled a thousand times and NTC ratio calculated as above. A NTC ratio less than zero indicates NTC under representation and a NTC ratio equal to zero means that no representation bias is observed.

## Statistics and data representation

P-values were calculated according to the indicated test. Error bars show standard deviations. ND for stands for Not Detected, NS for Not Significant, * $p < 0.05$, ** $p < 0.01$, *** $p < 0.001$, **** $p < 0.0001$. The experiments were replicated at least three times and representative images are shown.

## Supporting information

**S1 Fig. A3s expression in WT- and modified- A549 and HBEC3-KT cells.** A549 cells were infected with HAdV-A12, -B3, -C2 or mock control at a MOI = 3 and analyzed 2 days post infection. HBEC3-KT were infected with HAdV-A12, -B3, -C2 or mock control at a MOI = 1 and analyzed 4 days post infection. A3s expression levels were also reported in uninfected A549 shA3B and HBEC3-KT A3B. (A) The A3 mRNAs were quantified by RT-qPCR and expressed relative the GAPDH. The A3A, A3D and A3H mRNAs were not detected. (B) The A3B protein levels were assessed by western blot using the 5210-87-13 mAb antibody, quantified by densitometry and normalized with HSP90. Panel B displays a full blot from 15kDa to 55 kDa.
(TIFF)

**S2 Fig. A3B slows down intracellular viral DNA production.** HBEC-WT, -GFP, -A3B and -A3B-DD were infected with HAdV-A12, -B3 or -C2 at a MOI = 0.03. Intracellular viral DNA levels were quantified by qPCR at 6- and 24-hours post infection and at 2-, 4- and 7-days post infection. The results for the three replicates are depicted.
(TIFF)

**S3 Fig. A3B slows down extracellular viral DNA production.** HBEC-WT, -GFP, -A3B and -A3B-DD were infected with HAdV-A12, -B3 or -C2 at a MOI = 0.03. Extracellular viral DNA levels were quantified by qPCR at 2-, 4- and 7-days post infection. The results for the three replicates are depicted.
(TIFF)

**S4 Fig. Hypermutated viral sequences encode mutant or truncated proteins.** HBEC-WT, -GFP, -A3B and -A3B-DD were infected with HAdV-A12, -B3 or -C2 and the intracellular DNA was extracted 48 hours post infection. 3DPCR reactions conducted on different viral genes were cloned and sequenced. Hypermutated sequences detected in A3B-expressing cells (A3B) were aligned against the reference viral genome (Ref). The first 100 base pairs of representative examples are depicted. Mutated bases are underlined. Synonymous substitutions are colored in green, non-synonymous are colored in red and Stop codon are symbolized by a dash highlighted in yellow.
(TIFF)

**S5 Fig. A3B knockdown promotes replication of strain A12.** A549-WT, -scramble and -shA3B were infected with HAdV-A12, -B3 or -C2 at a MOI = 0.03. The percentage of infected cells were quantified by flow cytometry at 2-, 4-, 7-days post infection (dpi). The results for the three replicates are depicted.
(TIFF)

**S6 Fig. A3B knockdown promotes replication of strain A12.** A549-WT, -scramble and -shA3B were infected with HAdV-A12, -B3 or -C2 at a MOI = 0.03. Intracellular viral DNA levels were quantified by qPCR at 6- and 24-hours post infection and at 2-, 4- and 7-days post infection. The results for the three replicates are depicted.
(TIFF)

**S7 Fig. A3B knockdown promotes replication of strain A12.** A549-WT, -scramble and -shA3B were infected with HAdV-A12, -B3 or -C2 at a MOI = 0.03. Extracellular viral DNA levels were quantified by qPCR at 2-, 4- and 7-days post infection. The results for the three replicates are depicted.
(TIFF)

**S8 Fig. Decreased deaminase activity in A12-infected cells despite a high level of A3B protein.** A549 cells were infected with HAdV-A12, -B3, -C2 or a mock control at MOI = 3. Cell extracts were performed after 1- and 2-days post-infection. A3B protein levels (X-axis) were plotted against cell lysate deaminase activities (Y-axis).
(TIFF)

**S9 Fig. A3B protein half-life is increased by infection with the A12 strain.** A549 cells were infected with HAdV-A12, -B3, -C2 or a mock control at MOI = 3. Cycloheximide was added at 16 hpi. The levels of A3B, p53 and HPS90 proteins were evaluated by western blot for a period of 8 hours.
(TIFF)

**S10 Fig. A3B recolocalizes inside the viral replication centers in A12-infected cells.** A549 WT or A549 shA3B were infected with HAdV-A12 at a MOI = 3 and analyzed 24 hours post infection. (A) The A3B protein (green) and the viral replication centers as identified by EdU labelling (red) were imaged by fluorescence microscopy. (B) A3B labelling was quantified within the EdU+ area. Each circle represents a given cell. P-values were calculated by unpaired t-tests.
(TIFF)

**S1 Text.** Table A. Primers for cellular and viral mRNAs quantification. Table B. Primers for adenovirus DNA quantification. Table C. Primers and cycling conditions for 3DPCR. Table D. Primers for the discrimination of the APOBEC3B mRNA isoforms.
(DOCX)

## Acknowledgments

We thank Prof. Jerry W. Shay for the HBEC3-KT cell line, Prof. Slobodan Paessler for the PKR-deficient A549 cell line, Prof. Lieve Naesens for the HAdV-B3 and -C2 isolates, Prof. Warner C. Greene for the pSicoR-MS2 plasmids and Prof. Reuben Harris for the pLenti4_A3B and pLenti4-A3B-E68Q-E255Q expression plasmid. The following reagent was obtained through the NIH AIDS Reagent Program, Division of AIDS, NIAID, NIH: Anti-Human APO-BEC3B Monoclonal (5210-87-13) from Prof. Reuben Harris (cat# 12397). We thank the UNamur MORPH-IM platform for flow cytometry and fluorescent microscopy imaging. We thank Prof. Henri-François Renard for his microscopy advice.

## Author Contributions

**Conceptualization:** Noémie Lejeune, Nicolas A. Gillet.

**Formal analysis:** Noémie Lejeune.

**Funding acquisition:** Nicolas A. Gillet.

**Investigation:** Noémie Lejeune, Sarah Mathieu, Alexandra Decloux, Florian Poulain, Zoé Blockx, Kyle A. Raymond, Kévin Willemart, Rodolphe Suspène.

**Methodology:** Jean-Pierre Vartanian, Rodolphe Suspène.

**Project administration:** Nicolas A. Gillet.

**Supervision:** Nicolas A. Gillet.

**Validation:** Noémie Lejeune.

**Visualization:** Noémie Lejeune, Rodolphe Suspène, Nicolas A. Gillet.

**Writing – original draft:** Noémie Lejeune, Nicolas A. Gillet.

**Writing – review & editing:** Noémie Lejeune, Kyle A. Raymond, Jean-Pierre Vartanian, Rodolphe Suspène, Nicolas A. Gillet.

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
