## [Decision Letter · Decision Letter 0]

24 Oct 2022

Dear Prof gillet,

Thank you very much for submitting your manuscript "The APOBEC3B cytidine deaminase is an adenovirus restriction factor" for consideration at PLOS Pathogens. As with all papers reviewed by the journal, your manuscript was reviewed by members of the editorial board and by several independent reviewers. In light of the reviews (below this email), we would like to invite the resubmission of a significantly-revised version that takes into account the reviewers' comments.

The reviewers brought up several points that would need to be addressed before your manuscript could be reconsidered. Most importantly, they would like mechanistic data for the reduction of ABOBEC3B levels.

We cannot make any decision about publication until we have seen the revised manuscript and your response to the reviewers' comments. Your revised manuscript is also likely to be sent to reviewers for further evaluation.

Sincerely,

Robert F. Kalejta

Associate Editor

PLOS Pathogens

Patrick Hearing

Section Editor

PLOS Pathogens

Kasturi Haldar

Editor-in-Chief

PLOS Pathogens

orcid.org/0000-0001-5065-158X

Michael Malim

Editor-in-Chief

PLOS Pathogens

orcid.org/0000-0002-7699-2064

The reviewers brought up several points that would need to be addressed before your manuscript could be reconsidered. Most importantly, they would like mechanistic data for the reduction of ABOBEC3B levels.

Reviewer's Responses to Questions

**Part I - Summary**

Reviewer #1: This manuscript by Lejeune and coworkers is the first to my knowledge to comprehensively assess the potential interaction between APOBEC3B (A3B) and adenoviruses. In general the work is well done, and I like the fact that most experiments involve 3 strains and biological/technical triplicates. The most important experiments show that A3B overexpression lowers AdV replication and increases the detection of hypermutants and A3B knockdown does the opposite. An intriguing aspect is that A-family AdV seem more susceptible than other families B and C also tested here. An unresolved question is how AdV protect themselves from completely lethal restriction. To date, all restriction-susceptible virus types have mechanisms to fight back (Vif for lentis, Bet for foamys, RNR for herpes, etc) but this can be a topic for future studies. My comments below are mostly to improve rigor and clarity.

Reviewer #2: The manuscript by Lejeune and colleagues provides evidence that the adenovirus genome is deaminated by APOBEC3B during virus replication. Their lab showed previously that adenovirus genomes bore signatures of APOBEC3 deamination and here they attempt to demonstrate this experimentally. Some of the data regarding the effects of over- and under-expression are not that convincing, in part because of the large variation between "biological" replicates. Nonetheless, it does appear that Ad12 may be more affected by APOBEC3B than are the B3 or C2 strains. The most interesting data in the manuscript is presented in Figure 7, where they show that APOBEC3B protein levels are decreased in cells infected with the B3 or C2 strains but not Ad12. If more data were presented about the mechanism by which this occurs, the significance and novelty of the manuscript would be greatly increased.

Reviewer #3: In this manuscript, Lejeune et al. found that A3B is a critical restriction factor suppressing adenovirus through a deaminase-dependent mechanism. The authors convincingly demonstrated that A3B prevents viral infection by directly inducing mutations in the virus genome leading to a decrease in viral replication and infectivity. Moreover, they found that certain adenovirus strains counteract A3B by inducing its degradation. The results presented in this study are very well executed without any major flaws.

**Part II – Major Issues: Key Experiments Required for Acceptance**

Reviewer #1: Major:

1) Controls in Fig S1 should be merged into Fig 1. Readers need to understand the experimental system (maybe with cartoon) and the controls that were done to help establish it.

2) What is the A3B overexpression level in the cells in Fig 1, and how do these relate to physiological levels in primary cell types/tissues normally infected by adenoviruses? Such comparative data would be a nice supplementary figure.

3) The controls in Fig S5 for knockdown experiment are also very important and should be incorporated into the main Fig 5. As for #1 above, how do A3B levels in this cancer cell line compare to those observed in vivo in cell types/tissues normally infected by adenovirus?

4) The A3B westerns in Fig S5 and Fig7A raise the question of specificity. The band shown could be A3G, which sometimes migrates around 37kDa. A3B invariably migrates faster than 37kDa. Please show a control RTqPCR for related A3 family members, which will help demonstrate specificity (perhaps A3G is not expressed in this system) and that the observed changes are A3B-specific as implied here (which will also support Fig5/S5 studies). The 5210-87-13 mAb used for these experiments from the Harris lab recognizes a c-terminal epitope in A3A, A3B, and A3G so showing the full blot from 15-50kDa will be additionally helpful for interpretation.

5) Mechanism of A3B counteraction by AdV unclear and may involve lowering A3B protein levels and/or deaminase activity. The authors begin to address this point in Fig 7. However, the experiments in Fig 7A and D should really be coupled --- ie. readers need to know both A3B protein levels and activity levels simultaneously, with appropriate controls. For instance, is the diminished activity in Fig 7D due to lower A3B protein levels or potentially to less extract in the deaminase activity reaction?

6) The colocalization experiment in Fig 8 is quite interesting. The authors should include their A3B KD/KO line as a specificity control (as well as RTqPCR data as requested above in comment #4).

7) The replication center section of the discussion is confusing and misleading. The authors should scrap comparisons with cancer mutation (where no one has “caught” A3B in action) and focus instead on comparisons with other viruses. For instance, herpesviruses relocalize A3B (ref13,14) and PyV strangely induces A3B but the induced protein does not appear to co-localize with T-Ag/EdU positive replication centers (ref16). The latter paper has images that are comparable/contrastable to those here.

8) The authors suggest “…that adenoviruses evolved different mechanisms to antagonize APOBEC3B.” Alternatively, one mechanism might exist with different efficiencies depending on adenovirus species. Another might be that different species have different rates of aborted replication intermediates and what you’re detecting by 3DPCR may never be capable of replication. Please include preferred interpretation, as well as alternative interpretations, in discussion.

Reviewer #2: 1. It is really not clear to me when during adenovirus replication the authors think APOBEC3B might be acting. APOBEC3 enzymes preferentially deaminate single-stranded DNA, but it was my impression that this doesn't really exist during adenovirus replication because the DNA BP coats the newly replicated ssDNA. Do the authors think that APOBEC3B displaces DBP?

2. Figure 3. Isn't 24hr post-infection too late to examine immediate early and early gene expression? Or does this represent virus spread? It seems to me that this expression should be examined earlier after infection. Also, do the authors have any explanation for as to why there is no difference in B3 penton transcripts, but there is a difference in infectious particle production?

3. Figures 4 and 6. The authors do report deamination of B3 and C2 genomes, both in over-expression and knockdown cells. Unfortunately, sequencing the products that amplify at lower temperatures doesn't allow one to look at differences in the extent of deamination between the different viruses, only the consensus sequences. It would be much more interesting to report whether the B3 and C2 genomes are less deaminated that Ad12, which would require a deep-sequencing approach. This would also support the data in Figure 7.

4. Figure 7. Interestingly, B3 seems to reduce APOBEC3B levels more than C2, yet B3 replication was more affected than C2. How do the authors explain this? In part D), do the authors know if any other APOBEC3s are expressed in these cells? In other words, is all the deaminase activity attributable to APOBEC3B? What happens to the activity in APOBEC3B knockdown cells?

Reviewer #3: My main criticism is there is no data supporting that the decrease of A3B protein level is mediated through a degradation process after B3 and C2 infection. Can the level of A3B be restored by blocking the proteasome, certain viral proteins, or specific ubiquitin ligase complexes?

**Part III – Minor Issues: Editorial and Data Presentation Modifications**

Reviewer #1: Minor:

9) Are most experiments here true biological replicates (different virus stocks/cells on different days), or a technical replicates from replica experiments run in parallel at same time with the same viral stocks/cells?

10) Why do the authors rely on a single A3B knockdown construct instead of a much cleaner crispr knockout, at least for further validation?

11) Mutation clusters should be called exactly that – strand-coordinated clusters (or kataegis); hypermutation/editing is more general, and as the authors are aware originally associated with hypermutated HIV-1 genomes (high almost everywhere and not necessarily clustered). Clustered changes are, by definition, a requirement for 3DPCR, as the primer pairs only sample very short DNA regions and lower-denaturation temp amplicons only appear if multiple G/C-to-A/T changes have lowered the overall H-bond potential…

12) That said, can anything be inferred based on the strand that is accumulating C-to-U changes, as all of the reported clusters appear to be strand-biased…? Is this the same strand of the AdV genome?

Reviewer #2: 1. Line 93. The TC motif is not favored by APOBEC3G. Although the authors mention this later on, it should be stated here.

2. I was unclear as to what the authors meant throughout the manuscript by "biological replicates". Were these technical replicates or experimental replicates? If the former, then experimental replicates are needed to do statistics.

Reviewer #3: - What is the expression level of the other APOBEC3 family members in A549 and HBEC cells? Could other APOBEC3s expressed in these cell lines also participate in adenovirus restriction?

- The deaminase assay presented in Fig7D and S5C have no negative control without cell extract. In addition, the difference in DNA deaminase activity between the mock and A12 could be better illustrated, for example with cell extract titration.

-Is the localization of A3B to the viral replication center mediated by the NTD or CTD domains that are both known to bind single-stranded DNA?

PLOS authors have the option to publish the peer review history of their article (what does this mean?). If published, this will include your full peer review and any attached files.

Reviewer #1: No

Reviewer #2: No

Reviewer #3: **Yes: **Rémi Buisson
---

## [Decision Letter · Decision Letter 1]

20 Jan 2023

Dear Prof gillet,

Thank you very much for submitting your manuscript "The APOBEC3B cytidine deaminase is an adenovirus restriction factor" for consideration at PLOS Pathogens. As with all papers reviewed by the journal, your manuscript was reviewed by members of the editorial board and by several independent reviewers. The reviewers appreciated the attention to an important topic. Based on the reviews, we are likely to accept this manuscript for publication, providing that you modify the manuscript according to the review recommendations.

The reviewers were happy with the modified manuscript. One reviewer has some lingering questions that we would like to give you the opportunity to address. This most likely can be done by simple additions to the text, although the inclusion of new data if it is available (specifically, if there are any data to address potential differential deamination of different genomic loci) is encouraged. A revised manuscript would be editorially reviewed.

Sincerely,

Robert F. Kalejta

Academic Editor

PLOS Pathogens

Patrick Hearing

Section Editor

PLOS Pathogens

Kasturi Haldar

Editor-in-Chief

PLOS Pathogens

orcid.org/0000-0001-5065-158X

Michael Malim

Editor-in-Chief

PLOS Pathogens

orcid.org/0000-0002-7699-2064

The reviewers were happy with the modified manuscript. One reviewer has some lingering questions that we would like to give you the opportunity to address. This most likely can be done by simple additions to the text, although the inclusion of new data if it is available (specifically, if there are any data to address potential differential deamination of different genomic loci) is encouraged. A revised manuscript would be editorially reviewed.

Reviewer Comments (if any, and for reference):

Reviewer's Responses to Questions

**Part I - Summary**

Reviewer #1: The authors have done a nice job with revisions and have addressed all of my comments.

Reviewer #2: The authors have provided some additional data to support their observations that A12 but not B3 or C2 are affected by A3B, with the latter 2 viruses contributing to A3B degradation by an unknown mechanism. There are still many unanswered questions regarding the significance of their findings in the context of adenovirus replication.

Reviewer #3: The authors have addressed all my comments.

**Part II – Major Issues: Key Experiments Required for Acceptance**

Reviewer #1: None

Reviewer #2: 1. When A3B would have the opportunity to deaminate DNA. To me, this is a critical question. The authors explanation would suggest that that A3B binds more tightly to ssDNA than does the DNA BP. This doesn’t seem likely to me.

2. Timing of effect. There is no difference between in IE or E gene expression at 12 hr post-infection, and the only difference was at 24 hr in E4orf6 and penton expression with A12. I’m not sure how to reconcile these results – if the effect is only seen after virus spread, there should also be differences E1A expression at 24 hr. Or is this because there are fewer GC pairs in the E1A gene, so it is less susceptible to deamination? Have the authors looked at deamination in regions of the Ad genome besides E4? I still think there is a lack of correlation between different sets of data (penton expression in B3 infected cells is not reduced, but infectious particles are reduced).

3. 3. The authors now suggest that B3 and C2 at least partially degrade A3B. I still feel that this part of the manuscript is the most interesting and novel yet is under-developed – is there a viral protein that is responsible for this? I’m still not clear as to what the authors think is happening with Ad12 (“deaminase activity in cells infected with A12 is strongly reduced despite a still high amount of A3B protein.”), given that A3B does affect its infection.

Reviewer #3: It would be important to mention that epoxomicin treatment does not fully restore A3B level after B3 infection suggesting that it may have an additional mechanism leading to the decrease of A3B protein level

**Part III – Minor Issues: Editorial and Data Presentation Modifications**

Reviewer #1: None

Reviewer #2: (No Response)

Reviewer #3: (No Response)

PLOS authors have the option to publish the peer review history of their article (what does this mean?). If published, this will include your full peer review and any attached files.

Reviewer #1: No

Reviewer #2: No

Reviewer #3: **Yes: **Rémi Buisson

Figure Files:

Data Requirements:

Reproducibility:

References:

---

## [Editor Report · Decision Letter 2]

26 Jan 2023

Dear Prof gillet,

We are pleased to inform you that your manuscript 'The APOBEC3B cytidine deaminase is an adenovirus restriction factor' has been provisionally accepted for publication in PLOS Pathogens.

Best regards,

Robert F. Kalejta

Academic Editor

PLOS Pathogens

Patrick Hearing

Section Editor

PLOS Pathogens

Kasturi Haldar

Editor-in-Chief

PLOS Pathogens

orcid.org/0000-0001-5065-158X

Michael Malim

Editor-in-Chief

PLOS Pathogens

orcid.org/0000-0002-7699-2064
---

## [Editor Report · Acceptance letter]

2 Feb 2023

Dear Prof gillet,

We are delighted to inform you that your manuscript, "The APOBEC3B cytidine deaminase is an adenovirus restriction factor," has been formally accepted for publication in PLOS Pathogens.

Best regards,

Kasturi Haldar

Editor-in-Chief

PLOS Pathogens

orcid.org/0000-0001-5065-158X

Michael Malim

Editor-in-Chief

PLOS Pathogens

orcid.org/0000-0002-7699-2064